# Hierarchical VAEs provide a normative account of motion processing in the primate brain

**Hadi Vafaii**[1]
vafaii@umd.edu

**Jacob L. Yates**[2]
yates@berkeley.edu

**Daniel A. Butts**[1]
dab@umd.edu

[1]University of Maryland, College Park    [2]UC Berkeley

## Abstract

The relationship between perception and inference, as postulated by Helmholtz in the 19th century, is paralleled in modern machine learning by generative models like Variational Autoencoders (VAEs) and their hierarchical variants. Here, we evaluate the role of hierarchical inference and its alignment with brain function in the domain of motion perception. We first introduce a novel synthetic data framework, Retinal Optic Flow Learning (ROFL), which enables control over motion statistics and their causes. We then present a new hierarchical VAE and test it against alternative models on two downstream tasks: (i) predicting ground truth causes of retinal optic flow (e.g., self-motion); and (ii) predicting the responses of neurons in the motion processing pathway of primates. We manipulate the model architectures (hierarchical versus non-hierarchical), loss functions, and the causal structure of the motion stimuli. We find that hierarchical latent structure in the model leads to several improvements. First, it improves the linear decodability of ground truth factors and does so in a sparse and disentangled manner. Second, our hierarchical VAE outperforms previous state-of-the-art models in predicting neuronal responses and exhibits sparse latent-to-neuron relationships. These results depend on the causal structure of the world, indicating that alignment between brains and artificial neural networks depends not only on architecture but also on matching ecologically relevant stimulus statistics. Taken together, our results suggest that hierarchical Bayesian inference underlines the brain's understanding of the world, and hierarchical VAEs can effectively model this understanding.

## 1 Introduction

Intelligent interactions with the world require representation of its underlying composition. This inferential process has long been postulated to underlie human perception [1–9], and is paralleled in modern machine learning by generative models [10–17], which learn latent representations of their sensory inputs. The question of what constitutes a "good" representation has no clear answer [18, 19], but several desirable features have been proposed. In the field of neuroscience, studies focused on object recognition have suggested that effective representations "*untangle*" the various factors of variation in the input, rendering them linearly decodable [20, 21]. This intuitive notion of linear decodability has emerged in the machine learning community under different names such as "*informativeness*" [22] or "*explicitness*" [23]. Additionally, it has been suggested that "*disentangled*" representations are desirable, wherein distinct, informative factors of variations in the data are separated [24–29]. Artificial neural networks (ANNs) are also increasingly evaluated based on their alignment with biological neural processing [30–38], because of the shared goals of ANNs and the brain's sensory processing [25, 39, 40]. Such alignment also provides the possibility of gaining insights into the brain by understanding the operations within an ANN [41–47].

37th Conference on Neural Information Processing Systems (NeurIPS 2023).

In this work, we investigate how the combination of (i) model architecture, (ii) loss function, and (iii) training dataset, affects learned representations, and whether this is related to the brain-alignment of the ANN [41, 44]. We focus specifically on understanding the representation of motion because large sections of the visual cortex are devoted to processing motion [34], and the causes of retinal motion (moving objects and self-motion [48]) can be manipulated systematically. Crucially, motion in an image can be described irrespective of the identity and specific visual features that are moving, just as the identity of objects is invariant to how they are moving. This separation of motion and object processing mirrors the division of primate visual processing into dorsal (motion) and ventral (object) streams [49–51].

We designed a *naturalistic* motion simulation based on distributions of ground truth factors corresponding to the location and depth of objects, motion of these objects, motion of the observer, and observer's direction of gaze (i.e., the fixation point; Fig. 1a). We then trained and evaluated an ensemble of autoencoder-based models using our simulated retinal flow data. We based our evaluation on (1) whether the models untangle and disentangle the ground truth factors in our simulation; and (2) the degree to which their latent spaces could be directly related to neural data recorded in the dorsal stream of primates (area MT).

We introduce a new hierarchical variational autoencoder, the "compressed" Nouveau VAE (cNVAE) [52]. The cNVAE exhibited superior performance compared to other models across our multiple evaluation metrics. First, it discovered latent factors that accurately captured the ground truth factors in the simulation in a more disentangled manner than other models. Second, it achieved significant improvements in predicting neural responses compared to the previous state-of-the-art model [34], doubling the performance, with sparse mapping from its latent space to neural responses.

Taken together, these observations demonstrate the power of the synthetic data framework and show that a single inductive bias—hierarchical latent structure—leads to many desirable features of representations, including brain alignment.

## 2   Background & Related Work

**Neuroscience and VAEs.**   It has long been argued that perception reflects unconscious inference of the structure of the world constructed from sensory inputs. The concept of "perception as unconscious inference" has existed since at least the 19th century [1, 2], and more recently inspired Mumford [3] to conjecture that brains engage in hierarchical Bayesian inference to comprehend the world [3, 4]. These ideas led to the development of Predictive Coding [5, 9, 53–58], Bayesian Brain Hypothesis [6, 59–65], and Analysis-by-Synthesis [7], collectively suggesting that brains contain an internal generative model of the world [7, 8, 66, 67]. A similar idea underlies modern generative models [15–17, 68–70], especially hierarchical variants of VAEs [52, 71–73].

The Nouveau VAE (NVAE) [52] and very deep VAE (vdvae) [71] demonstrated that deep hierarchical VAEs can generate realistic high-resolution images, overcoming the limitations of their non-hierarchical predecessors. However, neither work evaluated how the hierarchical latent structure changed the quality of learned representations. Additionally, both NVAE and vdvae have an undesirable property: their convolutional latents result in a latent space that is several orders of magnitude larger than the input space, defeating a main purpose of autoencoders: compression. Indeed, Hazami et al. [74] showed that a tiny subset (around 3%) of the vdvae latent space is sufficient for comparable input reconstruction. Here, we demonstrate that it is possible to compress hierarchical VAEs and focus on investigating their latent representations with applications to neuroscience data.

**Evaluating ANNs on predicting biological neurons.**   Several studies have focused on evaluating ANNs on their performance in predicting brain responses, but almost entirely on describing static ("ventral stream") image processing [30, 33, 36]. In contrast, motion processing (corresponding to the dorsal stream) has only been considered thus far in Mineault et al. [34], who used a 3D ResNet ("DorsalNet") to extract ground truth factors about self-motion from drone footage ("AirSim", [75]) in a supervised manner. DorsalNet learned representations with receptive fields that matched known features of the primate dorsal stream and achieved state-of-the-art on predicting neural responses on the dataset that we consider here. In addition to our model architecture and training set, a fundamental difference between our approach and Mineault et al. [34] is that they trained their models using direct supervision. As such, their models have access to the ground truth factors at all times.

Table 1: ROFL categories used in this paper. ground truth factors include fixation point ($+2$); velocity of the observer when self-motion is present ($+3$); and, object position & velocity ($+6$). Figure 1b showcases a few example frames for each category. The stimuli can be rendered at any given spatial scale $N$, yielding an input shape of $2 \times N \times N$. Here we work with $N = 17$.

| Category | Description | Dimensionality |
|---|---|---|
| `fixate-1` | A moving observer maintains fixation on a background point. In addition, the scene contains one independently moving object. | $11 = 2 + 3 + 6$ |
| `fixate-0` | Same as `fixate-1` but without the object. | $5 = 2 + 3$ |
| `obj-1` | A single moving object, stationary observer. | $8 = 2 + 6$ |

Here, we demonstrate that it is possible to obtain ground truth factors "for free", in a completely unsupervised manner, while achieving better performance in predicting biological neuronal responses.

**Using synthetic data to train ANNs.** A core component of a reductionist approach to studying the brain is to characterize neurons based on their selectivity to a particular subset of pre-identified visual "features", usually by presenting sets of "feature-isolating" stimuli [76]. In the extreme, stimuli are designed that remove all other features except the one under investigation [77]. While these approaches can inform how pre-selected feature sets are represented by neural networks, it is often difficult to generalize this understanding to more natural stimuli, which are not necessarily well-described by any one feature set. As a result, here we generate synthetic data representing a *naturalistic* distribution of natural motion stimuli. Such synthetic datasets allow us to manipulate the causal structure of the world, in order to make hypotheses about what aspects of the world matter for the representations learned by brains and ANNs [78]. Like previous work on synthesized textures [15], here we specifically manipulate the data generative structure to contain factors of variation due to known ground truth factors.

## 3 Approach: Data & Models

**Retinal Optic Flow Learning (ROFL).** Our synthetic dataset framework, ROFL, generates the resulting optic flow from different world structures, self-motion trajectories, and object motion (Fig. 1a, see also [79]).

ROFL can be used to generate *naturalistic* flow fields that share key elements with those experienced in navigation through 3-D environments. Specifically, each frame contains global patterns that are due to self-motion, including rotation that can arise due to eye or head movement [80, 81]. In addition, local motion patterns can be present due to objects that move independently of the observer [48]. The overall flow pattern is also affected by the observer's direction of gaze (fixation point [82], Fig. 1a).

ROFL generates flow vectors that are instantaneous in time, representing the velocity across the visual field resulting from the spatial configuration of the scene and motion vectors of self and object. Ignoring the time-evolution of a given scene (which can arguably be considered separably [83]) dramatically reduces the input space from $[3 \times H \times W \times T]$ to $[2 \times H \times W]$, and allows a broader sampling of configurations without introducing changes in luminance and texture. As a result, we can explore the role of different causal structures in representation learning in ANNs.

The retinal flow patterns generated by a moving object depend on both the observer's self-motion and the rotation of their eyes as they maintain fixation in the world, in addition to the motion of the object itself. For example, Fig. 1c demonstrates a situation where the observer is moving forward, and the object is moving to the right, with different object positions: an object on the left side will have its flow patterns distorted, while an object on the right will have its flow patterns largely unaffected because its flow vectors are parallel with that of the self-motion. In summary, ROFL allows us to simulate retinal optic flow with a known ground truth structure driven by object and self-motion.

**The compressed NVAE (cNVAE).** The latent space of the NVAE is partitioned into groups, $z = \{z_1, z_2, \ldots, z_L\}$, where $L$ is the number of groups. The latent groups are serially dependent, meaning that the distribution of a given latent group depends on the value of the preceding latents,

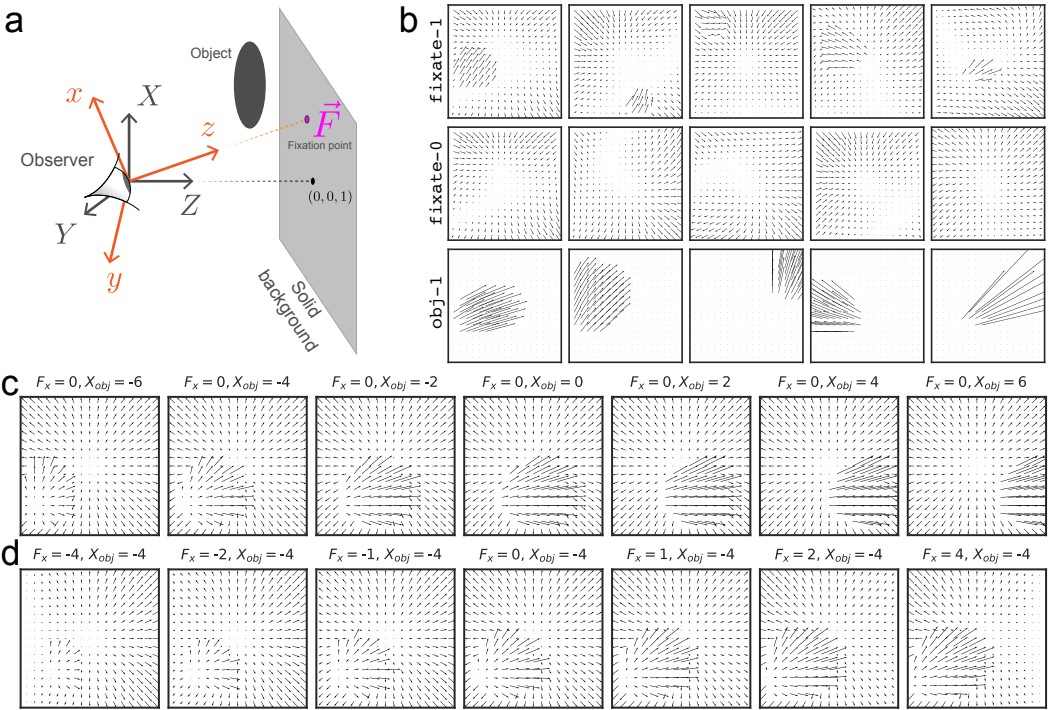

Figure 1: Retinal Optic Flow Learning (ROFL): a simulation platform for synthesizing naturalistic optic flow patterns. **(a)** The general setup includes a moving or stationary observer and a solid background, with optional moving object(s) in the scene. More details are provided in the appendix (section 13). **(b)** Example frames showcasing different categories (see Table 1 for definitions). **(c, d)** Demonstrating the causal effects of varying a single ground truth factor while keeping all others fixed: **(c)** $X_{obj}$, the $x$ component of object position (measured in retinal coordinates, orange), and **(d)** $F_x$, the $X$ component of the fixation point (measured in fixed coordinates, gray).

such that the prior is given by $p(\boldsymbol{z}) = p(\boldsymbol{z}_1) \cdot \prod_{\ell=2}^{L} p(\boldsymbol{z}_\ell|\boldsymbol{z}_{<\ell})$, and approximate posterior is given by $q(\boldsymbol{z}|\boldsymbol{x}) = \prod_{\ell=1}^{L} q(\boldsymbol{z}_\ell|\boldsymbol{z}_{<\ell}, \boldsymbol{x})$ (more details in section 9.1). Additionally, different latent groups in the NVAE operate at different spatial scales (Fig. 2, left), with multiple groups per scale. Crucially, such scale-dependent grouping is absent from non-hierarchical VAEs (Fig. 2, right).

The cNVAE closely follows the NVAE [52], with one important difference: the original NVAE latent space is convolutional, and ours is not. We modified the *sampler* layers (grey trapezoids, Fig. 2) such that their receptive field sizes match the spatial scale they operate on. Thus, sampler layers integrate over spatial information before sampling from the approximate posterior. The spatial patterns of each latent dimension are then determined by *expand* modules (yellow trapezoids, Fig. 2), based on a deconvolution step. Further details about the processing of the sampler and expand layers are provided in Supplementary section 9.2.

Our modification of the NVAE serves two purposes. First, it decouples spatial information from the functionality of latent variables, allowing them to capture abstract features that are invariant to particular spatial locations. Second, it has the effect of compressing the input space into a lower-dimensional latent code. We explain this in more detail in Supplementary section 9.3.

Our model has the following structure: 3 latent groups operating at the scale of $2 \times 2$; 6 groups at the scale of $4 \times 4$; and 12 groups at the scale of $8 \times 8$ (Table 4, Fig. 2). Therefore, the model has $3 + 6 + 12 = 21$ hierarchical latent groups in total. Each latent group has 20 latent variables, which results in an overall latent dimensionality of $21 \times 20 = 420$. See Table 4 and Supplementary section 9.3 for more details.

**Alternative models.** We evaluated a range of unsupervised models alongside cNVAE, including standard (non-hierarchical) VAEs [11, 12], a hierarchical autoencoder with identical architecture as

Table 2: Model details. Here, *hierarchical* means that there are parallel pathways for information to flow from the encoder to the decoder (Fig. 2), which is slightly different from the conventional notion. For variational models, this implies hierarchical dependencies between latents in a statistical sense [71]. This hierarchical dependence is reflected in the KL term for the cNVAE, where $L$ is the number of hierarchical latent groups. See Supplementary section 9.3 for more details and section 9.1 for a derivation. All models have an equal # of latent dimensions (420, see Table 4), approximately the same # of convolutional layers, and # of parameters ($\sim 24\ M$). EPE, endpoint error.

| Model | Architecture | Loss | Kullback–Leibler term (KL) |
|---|---|---|---|
| cNVAE | Hierarchical | EPE $+\beta * \mathrm{KL}$ | $\mathrm{KL} = \sum_{\ell=1}^{L} \mathbb{E}_{q(\boldsymbol{z}_{<\ell}\|\boldsymbol{x})}\big[\mathrm{KL}_\ell\big]$, where $\mathrm{KL}_\ell \coloneqq \mathcal{D}_{\mathrm{KL}}\big[q(\boldsymbol{z}_\ell\|\boldsymbol{x}, \boldsymbol{z}_{<\ell}) \, \| \, p(\boldsymbol{z}_\ell\|\boldsymbol{z}_{<\ell})\big]$ |
| VAE | Non-hierarchical | EPE $+\beta * \mathrm{KL}$ | $\mathrm{KL} = \mathcal{D}_{\mathrm{KL}}\big[q(\boldsymbol{z}\|\boldsymbol{x}) \, \| \, p(\boldsymbol{z})\big]$ |
| cNAE | Hierarchical | EPE | - |
| AE | Non-hierarchical | EPE | - |

the cNVAE but trained only with reconstruction loss (cNAE), and an autoencoder (AE) counterpart for the VAE (Table 2). All models had the same latent dimensionality (Table 4), and approximately the same number of parameters and convolutional layers. We used endpoint error as our measure of reconstruction loss, which is the Euclidean norm of the difference between actual and reconstructed flow vectors. This metric works well with optical flow data [84].

**Model representations.** We define a model's internal representation to be either the mean of each Gaussian for variational models (i.e., samples drawn from $q(\boldsymbol{z}|\boldsymbol{x})$ at zero temperature), or the bottleneck activations for autoencoders. For hierarchical models (cNVAE, cNAE), we concatenate representations across all levels (Table 4).

**Training details.** Models were trained for 160,000 steps at an input scale of $17 \times 17$, requiring slightly over a day on Quadro RTX 5000 GPUs. Please refer to Supplementary section 9.4 for additional details.

**Disentanglement and $\beta$-VAEs.** A critical decision when optimizing VAEs involves determining the weight assigned to the KL term in the loss function compared to the reconstruction loss. Prior research has demonstrated that modifying a single parameter, denoted as $\beta$, which scales the KL term, can lead to the emergence of disentangled representations [85, 86]. Most studies employing VAEs for image reconstruction typically optimize the standard evidence lower bound (ELBO) loss, where $\beta$ is fixed at a value of 1 [11, 52, 71]. However, it should be noted that due to the dependence of the reconstruction loss on the input size, any changes in the dimensionality of the input will inevitably alter the relative contribution of the KL term, and thus the "effective" $\beta$ [85].

Furthermore, Higgins et al. [16] recently established a strong correspondence between the generative factors discovered by $\beta$-VAEs and the factors encoded by inferotemporal (IT) neurons in the primate ventral stream. The

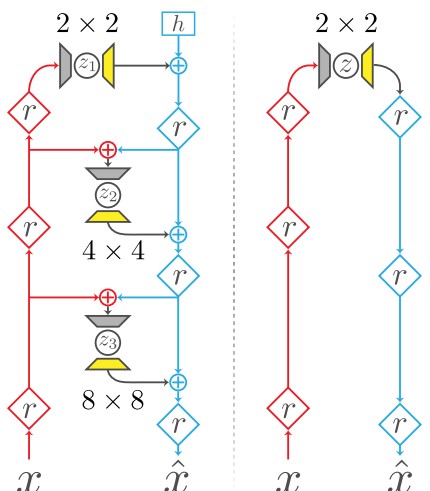

Figure 2: Architecture comparison. Left, compressed NVAE (cNVAE); right, non-hierarchical VAE. We modified the NVAE *sampler* layer (grey trapezoids) and introduced a deconvolution *expand* layer (yellow trapezoids). The encoder (inference) and decoder (generation) pathways are depicted in red and blue, respectively. $r$, residual block; $h$, trainable parameter; $+$, feature combination.

alignment between these factors and IT neurons exhibited a linear relationship with the value of $\beta$. In light of these findings, we explicitly manipulate the parameter $\beta$ within a range spanning from 0.01 to 10 to investigate the extent to which our results depend on its value.

# 4 Results

Our approach is based on the premise that the visual world contains a hierarchical structure. We use a simulation containing a hierarchical structure (ROFL, described above) and a hierarchical VAE (the cNVAE, above) to investigate how these choices affect the learned latent representations. While we are using a relatively simple simulation generated from a small number of ground truth factors, $g$, we do not specify how $g$ should be represented in our model or include $g$ in the loss. Rather, we allow the model to develop its own latent representation in a purely unsupervised manner. See Supplementary section 9.6 for more details on our approach.

We first consider hierarchical and non-hierarchical VAEs trained on the `fixate-1` condition (see Table 1; throughout this work, `fixate-1` is used unless stated otherwise). We extracted latent representations from each model and estimated the mutual information (MI) between the representations and ground truth factors such as self-motion, etc. For `fixate-1`, each data sample is uniquely determined using 11 ground truth factors (Table 1), and the models have latent dimensionality of 420 (Table 4). Thus, the resulting MI matrix has shape $11 \times 420$, where each entry shows how much information is contained in that latent variable about a given ground truth factor.

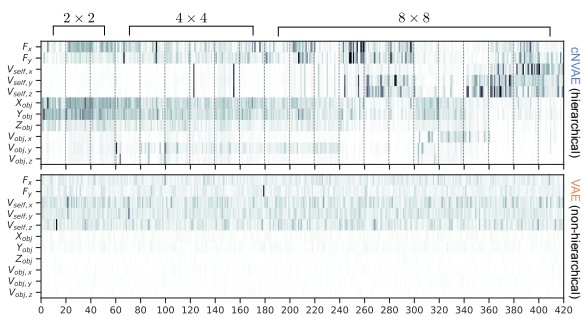

Figure 3: Mutual information between latent variables (x-axis) and ground truth factors (y-axis) is shown for cNVAE (top) and VAE (bottom). Dashed lines indicate 21 hierarchical latent groups of 20 latents each, comprising a 420-dimensional latent space. These groups operate at three different spatial scales, as indicated. In contrast, the VAE latent space lacks such grouping and operates solely at the spatial scale of $2 \times 2$ (see Fig. 2 and Table 4 for details on model latent configurations).

**Functional specialization emerges in the cNVAE.** Figure 3 shows the MI matrix for the latent space of cNVAE (top) and VAE (bottom). While both models achieved a good reconstruction of validation data (Fig. 14), the MI matrix for cN-VAE exhibits clusters corresponding to distinct ground truth factors at different levels of the hierarchy. Specifically, object-related factors of variation are largely captured at the top $2 \times 2$ scale, while information about fixation point can be found across the hierarchy, and self-motion is largely captured by $8 \times 8$ latent groups. In contrast, non-hierarchical VAE has no such structure, suggesting that the inductive bias of hierarchy enhances the quality of latent spaces, which we quantify next.

**Evaluating the latent code.** To demonstrate the relationship between ground truth factors and latent representations discovered by the cNVAE visible in Fig. 3, we apply metrics referred to as "untangling" and "disentengling". Additionally, in a separate set of experiments, we also evaluate model representations by relating them to MT neuron responses, which we call "brain-alignment". We discuss each of these in detail in the following sections.

**Untangling: the cNVAE untangles factors of variation.** One desirable feature of a latent representation is whether it makes information about ground truth factors easily (linearly) decodable [20, 21, 87]. This concept has been introduced in the context of core object recognition as "*untangling*". Information about object identity that is "tangled" in the retinal input is untangled through successive nonlinear transforms, thus making it linearly available for higher brain regions to extract [20]. This concept is closely related to the "*informativeness*" metric of Eastwood and Williams [22] and "*explicitness*" metric of Ridgeway and Mozer [23].

To assess the performance of our models, we evaluated the linear decodability of the ground truth factors, $g$, from model latent codes, $z$. Based on the $R^2$ scores obtained by predicting $g$ from $z$ using linear regression (Fig. 4), the cNVAE greatly outperforms competing models, faithfully capturing all ground truth factors. In contrast, the non-hierarchical VAE fails to capture object-related variables. Notably, the cNVAE can recover the fixation point location ($F_X$, $F_Y$) in physical space almost

perfectly. The fixation location has a highly nontrivial effect on the flow patterns, and varying it causes both global and local changes in the flow patterns (Fig. 1d).

Furthermore, cNVAE is the only model that reliably captures object position and velocity: especially note $V_{obj,z}$ (last column in Fig. 4). Inferring object motion from complex optic flow patterns involves two key components. First, the model must extract self-motion from flow patterns. Second, the model must understand how self-motion influences flow patterns globally. Only then can the model subtract self-motion from global flow vectors to obtain object motion. In vision science, this is known as the "*flow-parsing hypothesis*" [88–91]. Such flow-parsing is achieved by the cNVAE but none of the other models. See Supplementary section 11 for further discussion of this result and its implications.

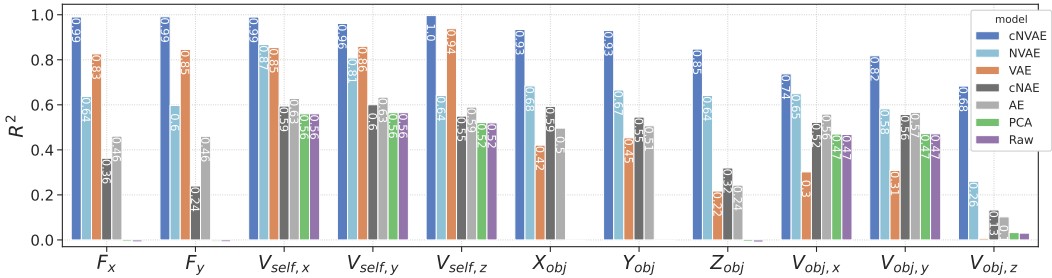

Figure 4: Hierarchical VAE untangles underlying factors of variation in data. The linear decodability of ground truth factors (x-axis) from different latent codes is shown. Untangling scores averaged across all ground truth factors are cNVAE = 0.898, NVAE = 0.639, VAE = 0.548, cNAE = 0.456, AE = 0.477, PCA = 0.236, and Raw = 0.235. For variational models, the best performing $\beta$ values were selected: cNVAE, $\beta = 0.15$; VAE, $\beta = 1.5$ (see Supplementary section 9.5 for more details).

**Disentanglement: the cNVAE produces more disentangled representations.** The pursuit of disentanglement in neural representations has garnered considerable attention [23, 85, 92–100]. In particular, Locatello et al. [19] established that learning fully disentangled representations is fundamentally impossible without inductive biases. Prior efforts such as $\beta$-VAE [85] demonstrated that increasing the weight of the KL loss (indicated by $\beta$) promotes disentanglement in VAEs. More recently, Whittington et al. [92] demonstrated that simple biologically inspired constraints such as non-negativity and energy efficiency encourage disentanglement. Here, we demonstrate that another biological inductive bias, hierarchy in the latent space, will promote disentanglement of the latent representations learned by VAEs.

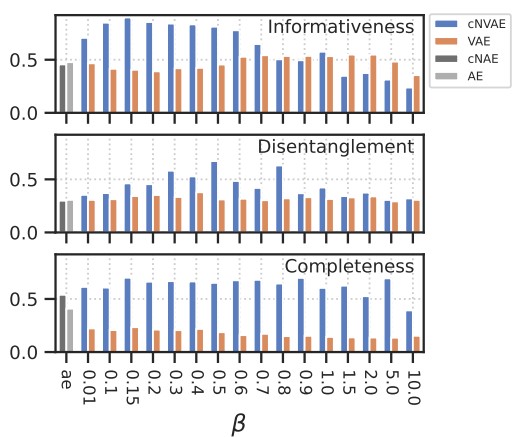

Figure 5: Evaluating the learned latent codes using the DCI framework [22]. Larger values are better for all metrics. Note that *informativeness* is closely related to *untangling* [20, 21]. See also Fig. 9.

To evaluate the role of hierarchy, we adopted the DCI framework [22] which offers a well-rounded evaluation of latent representations. The approach involves training a simple decoder (e.g., lasso regression) that predicts data generative factors $g$ from a latent code $z$; followed by computing a matrix of relative importances (e.g., based on lasso weights) which is then used to evaluate different aspects of the code quality: *Informativeness*—measures whether $z$ contains easily accessible information about $g$ (similar to untangling from above). *Disentanglement*—measures whether individual latents correspond to individual generative factors. *Completeness*—measures how many $z_i$ are required to capture any single $g_j$. If a single latent contributes to $g_j$'s prediction, the score will be 1 (complete). If all latent variables equally contribute to $g_j$'s prediction, the score will be 0 (maximally overcomplete). Note that "*completeness*" is also referred to as "*compactness*" [23]. See Fig. 9 and Supplementary

section 9.7.1 for more details, ref. [101] for a review, and ref. [102] for a recent extension of the DCI framework.

We follow the methods outlined by Eastwood and Williams [22] with two modifications: (1) we replaced lasso with linear regression to avoid the strong dependence on the lasso coefficient that we observed, and (2) we estimate the matrix of relative importances using a feature permutation-based algorithm (sklearn.inspection.permutation_importance), which measures the relative performance drop that results from shuffling a given latent.

We found that cNVAE outperforms competing models across all metrics for a broad range of $\beta$ values (Fig. 5). The observed pattern of an inverted U shape is consistent with previous work [85], which suggests that there is an optimal $\beta$ that can be empirically determined. In this case, cNVAE with $\beta = 0.5$ achieved the best average DCI score. Further, we found that VAEs lacking hierarchical structure learn highly overcomplete codes, such that many latents contribute to predicting a single ground truth factor. In conclusion, the simple inductive bias of hierarchy in the latent space led to a substantial improvement in VAE performance across all components of the DCI metric.

**Brain-alignment: the cNVAE aligns more closely with MT neurons.**    To evaluate the performance of models in predicting neuronal activity in response to motion stimuli, we used an existing dataset of $N = 141$ MT neurons recorded while presented with random dot kinematograms representing smoothly changing combinations of optic flow velocity fields [103, 104]. A subset of these neurons ($N = 84$) are publicly available on crcns.org, and were recently used in Mineault et al. [34] that we compare to.

To measure neuronal alignment, we first determined the mapping from each model's latent representation to MT neuron responses (binned spike counts, Fig. 6a). Here, the latent representation is defined as the mean of predicted Gaussian distributions for VAEs, and the bottleneck activations for AEs. We learn this linear latent-to-neuron mapping using ridge regression. Figure 6b shows the average firing rate of an example neuron along with model predictions. Because sensory neurons have a nonzero response latency, we determined each neuron's optimal response latency, which maximized cross-validated performance. The resulting distribution of best-selected latencies (Fig. 6c) peaked around $100 \ ms$: consistent with known MT latencies [103]. We also empirically optimized ridge coefficients to ensure each neuron has its best fit. Figure 6d shows that the models capture the receptive field properties of MT neurons as measured by the spike-triggered average stimulus. To evaluate performance, we follow methods established by Mineault et al. [34]: whenever repeated trials were available, we report Pearson's $R$ on that held-out data, normalized by maximum explainable variance [105]. When repeats were not available, we performed 5-fold cross-validation and reported the held-out performance using Pearson's $R$ between model prediction and spike trains.

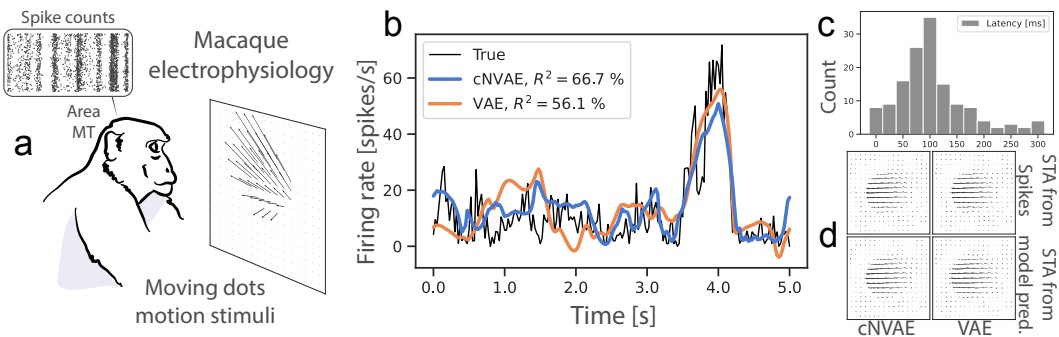

Figure 6: **(a)** Experimental setup form [103, 104]. **(b)** Both models explain MT neural variability well. **(c)** Distribution of best estimated latencies. **(d)** Spike-triggered averages (STA) are shown.

**Evaluating brain alignment.**    We use two measures of brain alignment: the success at predicting the neural response (Pearson's $R$, Fig. 7, Table 3); and, the "*alignment*" between neurons and individual model latents (Fig. 8, [16]). These mirror the untangling and completeness metrics described above (more details are provided below).

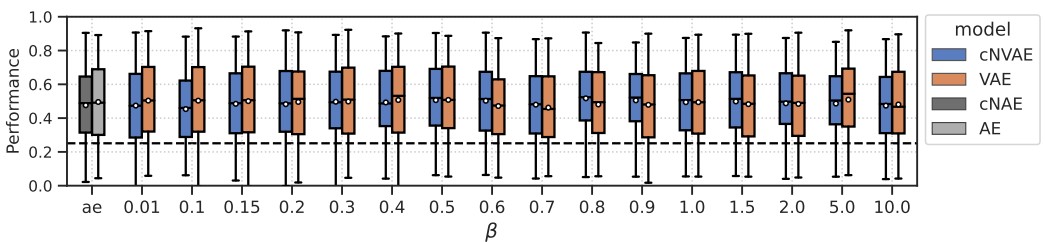

Figure 7: All models (pretrained on `fixate-1`) perform comparably in predicting MT neuron responses. Dashed line corresponds to the previous state-of-the-art on this data [106].

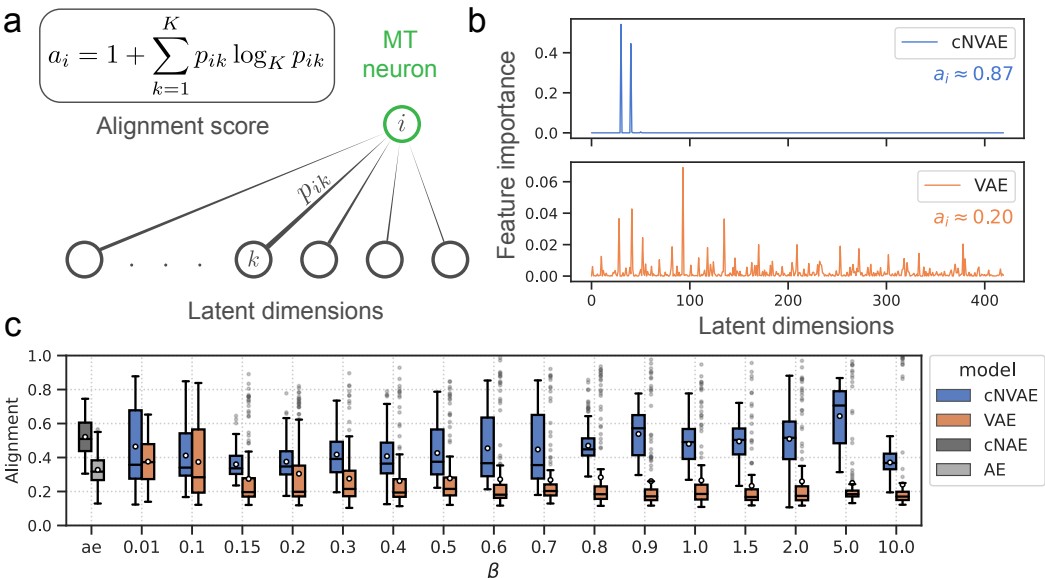

Figure 8: Hierarchical models (cNVAE, cNAE) are more aligned with MT neurons since they enable sparse latent-to-neuron relationships. **(a)** Alignment score measures the sparsity of permutation feature importances. $a_i = 0$ when all latents are equally important in predicting neuron $i$; and, $a_i = 1$ when a single latent predicts the neuron. **(b)** Feature importances are plotted for an example neuron (same as in Fig. 6b). cNVAE ($\beta = 0.01$) predicts this neuron's response in a much sparser manner compared to non-hierarchical VAE ($\beta = 5$). Supplementary section 9.5 contains a discussion of our rationale in choosing these $\beta$ values. **(c)** Alignment across $\beta$ values, and autoencoders (ae).

**All models predict MT neuron responses well.** After training a large ensemble of unsupervised models on `fixate-1` and learning the neural mapping, we found that both hierarchical (cNVAE & cNAE) and non-hierarchical (VAE & AE) variants had similar ability to predict neural responses (Fig. 7). The performance did depend on the loss function itself, with the variational loss outperforming simple autoencoder reconstruction loss (Table 3).

**Hierarchical VAEs are more aligned with MT neurons.** We next tested how these factors affect neural alignment, i.e., how closely neurons are related to individual latents in the model. Figure 8a demonstrates what we mean by "alignment": a sparse latent-to-neuron relationship means larger alignment, indicative of a similar representational "form" [16]. See Fig. 10 for an illustration of this idea. To formalize this notion, we use feature permutation importance (described above), applied to the ridge regression models. This yields a $420$-dimensional vector per neuron. Each dimension of this vector captures the importance of a given latent variable in predicting the responses of the neuron. We normalize these vectors and interpret them as the probability of importance. We then define alignment score $a_i$ of neuron $i$ as $a_i = 1 + \sum_{k=1}^{K} p_{ik} \log_K p_{ik}$, where $p_{ik}$ is interpreted as the importance of $k-$th latent variable in predicting neuron $i$ (Fig. 8a). This concept is closely related to the "*completeness*" score from the DCI framework as discussed above.

Table 3: Both cNVAE and VAE perform well in predicting MT neuron responses, surpassing previous state-of-the-art models by more than a twofold improvement. Moreover, the clear gap between `fixate-1` and other categories highlights the importance of pretraining data [107].

| Model | Pretraining dataset | Performance, $R$ ($\mu \pm se$; $N = 141$) | | | |
|---|---|---|---|---|---|
| | | $\beta = 0.5$ | $\beta = 0.8$ | $\beta = 1$ | $\beta = 5$ |
| cNVAE | `fixate-1` | $\mathbf{.506 \pm .018}$ | $\mathbf{.517 \pm .017}$ | $.494 \pm .018$ | $.486 \pm .016$ |
| | `fixate-0` | $.428 \pm .018$ | $.450 \pm .019$ | $.442 \pm .019$ | $.469 \pm .018$ |
| | `obj-1` | $.471 \pm .018$ | $.465 \pm .018$ | $.477 \pm .017$ | $.468 \pm .018$ |
| VAE | `fixate-1` | $\mathbf{.508 \pm .019}$ | $.481 \pm .018$ | $.494 \pm .018$ | $\mathbf{.509 \pm .018}$ |
| cNAE | `fixate-1` | $.476 \pm .018$ | | | |
| AE | `fixate-1` | $.495 \pm .019$ | | | |
| CPC [108] | AirSim [75] | $.250 \pm .020$ (Mineault et al. [34]) | | | |
| DorsalNet | AirSim [75] | $.251 \pm .019$ (Mineault et al. [34]) | | | |

For almost all $\beta$ values, the cNVAE exhibited a greater brain alignment than non-hierarchical VAE (Fig. 8c; cNVAE > VAE, paired $t-$test; see Fig. 16 and Table 5). Similarly, for the autoencoders, we found that the hierarchical variant outperformed the non-hierarchical one (cNAE > AE). Based on these observations, we conclude that higher brain alignment is primarily due to hierarchical latent structure. However, note that hierarchy in the traditional sense did not matter: all these models had approximately the same number of convolutional layers and parameters.

**Factors leading to brain-alignment.** To test the effect of the training dataset (i.e., category of ROFL) on model performance, we trained cNVAE models using `fixate-0`, `fixate-1`, and `obj-1` categories (Table 1), while also exploring a variety of $\beta$ values. We found that `fixate-1` clearly outperformed the other two ROFL categories (Table 3), suggesting that both global (e.g., self-motion) and local (e.g., object motion) sources of variation are necessary for learning MT-like representations. The effect of loss function was also visible: some $\beta$ values led to more alignment. But this effect was small compared to the effect of hierarchical architecture (Fig. 8c).

## 5   Discussion

We introduced a new framework for understanding and evaluating the representation of visual motion learned by artificial and biological neural networks. This framework provides a way to manipulate causes in the world and evaluate whether learned representations untangle and disentangle those causes. In particular, our framework makes it possible to test the influence of architecture (Fig. 2), loss function (Table 2), and training set (Table 1) on the learned representations, encompassing 3 out of the 4 core components of a recently proposed neuroconnectionist research programme [41]. Our framework brings hypothesis-testing to understand [biological] neural processing of vision and provides an interpretive framework to understand neurophysiological data.

The goal of the present work was to establish our framework and demonstrate its potential. To this end, we made several simplifying choices, such as training on individual flow frames rather than time-evolving videos. We provide a detailed discussion of study limitations in Supplementary section 8. Future work will address these by rendering images in simulations and using image-computable models, incorporating real eye-tracking and scene data in ROFL [83, 109], testing our approach on more data from other brain areas such as MST [110, 111], and using more sophisticated methods to measure representational alignment between ANNs and brains [112–115].

**Conclusion.** We used synthetic data to test how causal structure in the world affects the representations learned by autoencoder-based models and evaluated the learned representations based on how they represent ground truth factors and how well they align with biological brains. We found that a single inductive bias, hierarchical latent structure, leads to desirable representations and increased brain alignment.

## 6 Code & Data

Our code and model checkpoints are available here: https://github.com/hadivafaii/ROFL-cNVAE.

## 7 Acknowledgments

This work was supported by NSF IIS-2113197 (HV and DAB), NSF DGE-1632976 (HV), and NIH R00EY032179 (JLY). We thank our anonymous reviewers for their helpful comments, and the developers of the software packages used in this project, including PyTorch [116], NumPy [117], SciPy [118], scikit-learn [119], pandas [120], matplotlib [121], and seaborn [122].

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

Supplementary material for:

# Hierarchical VAEs provide a normative account of motion processing in the primate brain

## 8   Study Limitations & Considerations

We established a synthetic data framework that allows hypothesis generation and testing for neural processing of motion. While our paper opens up many interesting venues for future exploration, it is necessarily simplified, as it focuses on establishing our framework and demonstrating its potential. As such, it currently has several limitations:

First, our simulation generates velocity fields, rather than full spatiotemporal movies, which allows our model to avoid mimicking the complexities of motion extracted by the early visual pathway [123, 124]. This strategy also allows for direct comparison with recorded neural data in MT and MST using random dot kinematograms [103, 104, 106]. However, a more complex model would be necessary to explain responses of neurons earlier in the visual pathway, such as V1, which would require a pixel-computable model as in previous work (e.g., DorsalNet [34]). Likewise, our $> 2\times$ improvement in performance in explaining MT neural data over DorsalNet is likely due in part to their network trained to perform this extraction from spatiotemporal movies, which was not necessarily equivalent to the random dot velocity fields used in the experiments. Thus, it is an open question whether a hierarchical VAE trained and tested on video stimuli would better align to neural data than the model from Mineault et al. [34]. Future work in this space will involve rendering images in simulations and using image-computable models for a fair comparison.

Second, we chose relatively simple environments and simple fixation rules to generate our optic flow fields and avoided the true complexity of 3-D natural environments and their interaction with eye movements and self-motion as has recently been measured [83, 109]. The simplification of our simulation still demonstrates the importance of including such elements in understanding neural representations and provides a framework for incorporating real eye-tracking and scene data [83, 109] into future work with ROFL.

Finally, we only tested neural alignment on one experimental paradigm using neurons in area MT, which leaves the question of whether this is a general principle of brain computation. Addressing this requires testing our approach on more data from other brain areas, such as MST. Based on previous work [106, 110], we expect that hierarchical computation is even more necessary for MST, which is an open question to address in future work.

**Interpreting brain-alignment.**   We measured the alignment between ANN models and MT neurons using both linear predictive power (Fig. 7) and an alternative measure of alignment that is sensitive to the sparsity of latent-to-neuron relationships ("alignment-score", Fig. 8a). Linear regression has been most commonly used to measure similarity, or alignment, between pairs of representations [30, 31, 34, 36], but often results in degenerate [30, 36, 38] and unreliable [125] measures of representational alignment. Consistent with this, our application of linear regression found that it was not effective in differentiating between models: although the cNVAE produced the single best model in terms of neuronal prediction, we found that both hierarchical and non-hierarchical VAEs performed similarly in predicting MT neuron responses (Fig. 7).

In contrast, the alignment score (Fig. 8a) was much more consistent in distinguishing between models (see Fig. 16), and revealed that hierarchical models (both cNVAE and cNAE) had significantly sparser latent-to-neuron relationships. The alignment score measures whether a model has learned a similar representational "form" to the brain, which would enable the sparsity of latent-to-neuron relationships. This concept is closely related to the "*completeness*" score from the DCI framework [22]. The alignment score shown in Fig. 8a was also used in Higgins et al. [16], although they used the magnitude of nonzero coefficients as their feature importance (under lasso regression). However, we note that this alignment score also has limitations, and for example, it is not a proper metric in the mathematical sense [112]. Future work will consider more sophisticated metrics for brain alignment [112–115].

# 9 Additional Methods

## 9.1 VAE loss derivation

Suppose some observed data $\boldsymbol{x}$ are sampled from a generative process as follows:

$$p(\boldsymbol{x}) = \int p(\boldsymbol{x}, \boldsymbol{z}) \, d\boldsymbol{z} = \int p(\boldsymbol{x}|\boldsymbol{z})p(\boldsymbol{z}) \, d\boldsymbol{z}, \tag{1}$$

where $\boldsymbol{z}$ are latent (or unobserved) variables. In this setting, it is interesting to ask which set of latents $\boldsymbol{z}$ are likely, given an observation $\boldsymbol{x}$. In other words, we are interested in posterior inference

$$p(\boldsymbol{z}|\boldsymbol{x}) \propto p(\boldsymbol{x}|\boldsymbol{z})p(\boldsymbol{z}). \tag{2}$$

The goal of VAEs is to approximate the (unknown) true posterior $p(\boldsymbol{z}|\boldsymbol{x})$ with a distribution $q(\boldsymbol{z}|\boldsymbol{x}; \theta)$, where $\theta$ are some free parameters to be learned. This goal is achieved by minimizing the Kullback-Leibler divergence between the approximate posterior $q$ and the true posterior $p$:

$$\text{Goal:} \quad \text{minimize} \quad \mathcal{D}_{\text{KL}}\Big[q(\boldsymbol{z}|\boldsymbol{x}; \theta) \, \big\| \, p(\boldsymbol{z}|\boldsymbol{x})\Big]. \tag{3}$$

This objective is intractable, but rearranging the terms leads to the following loss that is also the (negative) variational lower bound on $\log p(\boldsymbol{x})$:

$$\mathcal{L}_{\text{VAE}} = -\mathbb{E}_q\Big[\log p(\boldsymbol{x}|\boldsymbol{z}; \boldsymbol{\theta}_{dec})\Big] + \mathcal{D}_{\text{KL}}\Big[q(\boldsymbol{z}|\boldsymbol{x}; \boldsymbol{\theta}_{enc}) \, \big\| \, p(\boldsymbol{z}|\boldsymbol{\theta}_{dec})\Big], \tag{4}$$

where $\boldsymbol{\theta}_{enc}$ and $\boldsymbol{\theta}_{dec}$ are the parameters for the encoder and decoder neural networks (see Fig. 2). The first term in Equation 4 is the reconstruction loss, which we chose to be the Euclidean norm of the differences between predicted and reconstructed velocity vectors (i.e., the endpoint error [84]). We will now focus on the second term in Equation 4, the KL term.

### 9.1.1 Prior, approximate posterior, and the KL term in vanilla VAE

In standard non-hierarchical VAEs, the prior is not parameterized. Instead, it is chosen to be a simple distribution such as a Gaussian with zero mean and unit covariance:

$$p(\boldsymbol{z}|\boldsymbol{\theta}_{dec}) \rightarrow p(\boldsymbol{z}) = \mathcal{N}(\boldsymbol{0}, \boldsymbol{I}). \tag{5}$$

The approximate posterior is also a Gaussian with mean $\boldsymbol{\mu}(\boldsymbol{x}; \boldsymbol{\theta}_{enc})$ and variance $\boldsymbol{\sigma}^2(\boldsymbol{x}; \boldsymbol{\theta}_{enc})$:

$$q(\boldsymbol{z}|\boldsymbol{x}; \boldsymbol{\theta}_{enc}) = \mathcal{N}(\boldsymbol{\mu}(\boldsymbol{x}; \boldsymbol{\theta}_{enc}), \boldsymbol{\sigma}^2(\boldsymbol{x}; \boldsymbol{\theta}_{enc})) \tag{6}$$

As a result, we see that the KL term for a vanilla VAE only depends on encoder parameters $\boldsymbol{\theta}_{enc}$.

### 9.1.2 Prior, approximate posterior, and the KL term in the cNVAE

Similar to the NVAE [52], the cNVAE latent space is organized hierarchically such that latent groups are sampled sequentially, starting from "top" latents all the way down to the "bottom" ones (i.e., $\boldsymbol{z}_1$ to $\boldsymbol{z}_3$ in Fig. 2). In addition to its hierarchical structure, another important difference between cNVAE and vanilla VAE is that the cNVAE prior is learned from data. Note the "mid" and "bottom" latent groups in Fig. 2, indicated as $\boldsymbol{z}_2$ and $\boldsymbol{z}_3$ respectively: the cNVAE is designed in a way that changing the parameters along the decoder pathway will impact the prior distributions on $\boldsymbol{z}_2$ and $\boldsymbol{z}_3$ (but not $\boldsymbol{z}_1$). Note the "$h$" in Fig. 2, which is also a learnable parameter. In summary, the "top" cNVAE latents (e.g., $\boldsymbol{z}_1$ in Fig. 2) have a fixed prior distribution similar to vanilla VAEs; whereas, the prior distribution for every other cNVAE latent group is parametrized and learned from data.

More formally, the cNVAE latents are partitioned into disjoint groups, $\boldsymbol{z} = \{\boldsymbol{z}_1, \boldsymbol{z}_2, \ldots, \boldsymbol{z}_L\}$, where $L$ is the number of groups. Then, the prior is represented by:

$$p(\boldsymbol{z}|\boldsymbol{\theta}_{dec}) = p(\boldsymbol{z}_1) \cdot \prod_{\ell=2}^{L} p(\boldsymbol{z}_\ell|\boldsymbol{z}_{<\ell}; \boldsymbol{\theta}_{dec}), \tag{7}$$

where the conditionals are represented by factorial Normal distributions. For the first latent group we have $p(\boldsymbol{z}_1) = \mathcal{N}(\boldsymbol{0}, \boldsymbol{I})$, similar to vanilla VAEs. For every other latent group, we have:

$$p(\boldsymbol{z}_\ell|\boldsymbol{z}_{<\ell}; \boldsymbol{\theta}_{dec}) = \mathcal{N}(\boldsymbol{\mu}(\boldsymbol{z}_{<\ell}; \boldsymbol{\theta}_{dec}), \boldsymbol{\sigma}^2(\boldsymbol{z}_{<\ell}; \boldsymbol{\theta}_{dec})), \tag{8}$$

where $\boldsymbol{\mu}(\boldsymbol{z}_{<\ell}; \boldsymbol{\theta}_{dec})$ and $\boldsymbol{\sigma}^2(\boldsymbol{z}_{<\ell}; \boldsymbol{\theta}_{dec})$ are outputted from the decoder *sampler* layers. Similarly, the approximate posterior in the cNVAE is represented by:

$$q(\boldsymbol{z}|\boldsymbol{x}; \boldsymbol{\theta}_{enc}) = \prod_{\ell=1}^{L} q(\boldsymbol{z}_\ell|\boldsymbol{z}_{<\ell}, \boldsymbol{x}; \boldsymbol{\theta}_{enc}). \tag{9}$$

We adopt a Gaussian parameterization for each conditional in the approximate posterior:

$$q(\boldsymbol{z}_\ell|\boldsymbol{z}_{<\ell}, \boldsymbol{x}; \boldsymbol{\theta}_{enc}) = \mathcal{N}(\boldsymbol{\mu}(\boldsymbol{z}_{<\ell}, x; \boldsymbol{\theta}_{enc}), \boldsymbol{\sigma}^2(\boldsymbol{z}_{<\ell}, \boldsymbol{x}; \boldsymbol{\theta}_{enc})), \tag{10}$$

where $\boldsymbol{\mu}(\boldsymbol{z}_{<\ell}, x; \boldsymbol{\theta}_{enc})$ and $\boldsymbol{\sigma}^2(\boldsymbol{z}_{<\ell}, \boldsymbol{x}; \boldsymbol{\theta}_{enc})$ are outputted from the encoder *sampler* layers (Fig. 2; grey trapezoids). We are now in a position to explicitly write down the KL term from Equation 4 for the cNVAE:

$$\text{KL term} = \mathcal{D}_{\text{KL}}\Big[q(\boldsymbol{z}_1|\boldsymbol{x}, \boldsymbol{\theta}_{enc}) \,\big\|\, p(\boldsymbol{z}_1)\Big] + \sum_{\ell=2}^{L} \mathbb{E}_{q(\boldsymbol{z}_{<\ell}|\boldsymbol{x}, \boldsymbol{\theta}_{enc})}\Big[\text{KL}_\ell(\boldsymbol{\theta}_{enc}, \boldsymbol{\theta}_{dec})\Big], \tag{11}$$

where $\text{KL}_\ell$ refers to the local KL term for group $\ell$ and is given by:

$$\text{KL}_\ell(\boldsymbol{\theta}_{enc}, \boldsymbol{\theta}_{dec}) := \mathcal{D}_{\text{KL}}\Big[q\left(\boldsymbol{z}_\ell|\boldsymbol{z}_{<\ell}, \boldsymbol{x}; \boldsymbol{\theta}_{enc}\right) \,\big\|\, p\left(\boldsymbol{z}_\ell|\boldsymbol{z}_{<\ell}, \boldsymbol{\theta}_{dec}\right)\Big], \tag{12}$$

and the approximate posterior up to the $(\ell-1)^{th}$ group is defined as:

$$q(\boldsymbol{z}_{<\ell}|\boldsymbol{x}; \boldsymbol{\theta}_{enc}) := \prod_{i=1}^{\ell-1} q(\boldsymbol{z}_i|\boldsymbol{z}_{<i}, \boldsymbol{x}; \boldsymbol{\theta}_{enc}). \tag{13}$$

This concludes our derivation of the KL terms shown in Table 2.

## 9.2 Key architectural differences between NVAE and cNVAE

Our main contribution to architecture design lies in the modification of the NVAE "*sampler*" layer and the introduction of the "*expand*" layer, which are simple deconvolution (Fig. 2). In the NVAE, the purpose of a sampler layer is to map encoder features to mean and variance vectors $\boldsymbol{\mu}(\boldsymbol{x})$ and $\boldsymbol{\sigma}^2(\boldsymbol{x})$, which are then used to construct approximate posterior distributions $q\left(\boldsymbol{z}|\boldsymbol{x}\right) = \mathcal{N}(\boldsymbol{\mu}(\boldsymbol{x}), \boldsymbol{\sigma}^2(\boldsymbol{x}))$. During the inference forward pass, downsampling operations will progressively reduce the spatial scale of representations along the encoder pathway, such that processed stimuli will have different spatial dimensions at different stages of inference. In the NVAE, all sampler layers are convolutional with a fixed kernel size of $3 \times 3$ and padding of 1. Thus, the application of a sampler layer does not alter the spatial dimension of hidden encoder features, resulting in a convolutional latent space.

For example, at level $\ell$ of the hierarchy, $\boldsymbol{\mu}_\ell(\boldsymbol{x})$ and $\boldsymbol{\sigma}_\ell^2(\boldsymbol{x})$ will have shapes like $d \times s_\ell \times s_\ell$, where $d$ is number of latent variables per latent group, and $s_\ell$ is the spatial scale of the processed encoder features at level $\ell$. As a result, NVAE latents themselves end up being convolutional in nature, with shapes like $d \times s_\ell \times s_\ell$. This design choice leads to the massive expansion of NVAE latent dimensionality, which is often orders of magnitude larger than the input dimensionality.

To address this, we modified NVAE sampler layers in a manner that would integrate over spatial information before the sampling step, achieving a compressed, non-convolutional, and more abstract latent space. Specifically, we designed the cNVAE sampler layers such that their kernel sizes are selected adaptively to match the spatial scale of their input feature. For example, if latent group $\ell$ operates at the scale of $s_\ell$, then the sampler layer will have a kernel size of $s_\ell \times s_\ell$, effectively integrating over space before the sampling step. This results in latents with shape $d \times 1 \times 1$.

Finally, combining the latents with decoder features requires projecting them back into convolutional space. We achieve this by introducing "*expand*" modules (Fig. 2; yellow trapezoids) which are simple deconvolution layers with a kernel size of $s_\ell \times s_\ell$. Thus, processing a $d \times 1 \times 1$ latent by an expand layer results in a $d \times s_\ell \times s_\ell$ output, allowing them to be combined with the decoder features (concatenation or addition; blue "+" in Fig. 2).

**cNVAE pros and cons.** Our approach resulted in a much smaller latent space compared to NVAE, which allowed us to examine the function of individual latents, decoupled from their spatial influence. However, one drawback of our approach is that it makes the architecture input-size-dependent. Scaling up the input increases the parameter count of cNVAE. This did not pose serious problems in the present work, as we generated ROFL datasets in a relatively small spatial scale of $17 \times 17$. However, this dependence on input dimensionality could cause problems when larger datasets are considered. To mitigate this, one possibility is to use depthwise separable convolutions for sampler and expand layers.

**Other details.** We used the same regular residual cells for both the encoder and the decoder ("*r*" in Fig. 2). We did not experience memory issues because of the relatively small scale of our experiments. However, rendering the input data at a larger spatial dimension would lead to larger memory usage, which might require using depthwise separable convolutions as in Figure 3b in Vahdat and Kautz [52]. Similar to NVAE, we found that the inclusion of squeeze & excitation [126] helped. Contrary to the suggestion of NVAE [52] and others [72], batch norm [127] destabilized training in our case. Instead, we found that weight normalization [128] and swish activation [129, 130] were both instrumental in training. We used weight normalization without data-dependent initialization. Our observations also deviate from those of Vahdat and Kautz [52] with regard to spectral regularization [131]. We achieved stable training and good results without the need for spectral regularization. This is perhaps because our dataset is synthetic and has a much smaller dimensionality compared to real-world datasets such as faces considered in the NVAE. Finally, we also found that residual Normal parameterization of the approximate posterior was helpful.

**Model hyperparameters.** The latent space of cNVAE contains a hierarchy of latent groups that are sampled sequentially: "top" latents are sampled first, all the way down to "bottom" latents that are closest to the stimulus space (Fig. 2). Child [71] demonstrated that model depth, in this statistical sense, was responsible for the success of hierarchical VAEs. Here, we increased the depth of cNVAE architecture while ensuring that training remained stable. Our final model architecture had a total of 21 hierarchical latent groups: 3 groups operating at the spatial scale of $2 \times 2$, 6 groups at the scale of $4 \times 4$, and 12 groups at the scale of $8 \times 8$ (see Figs. 2 and 3). Each latent group has 20 latents, which yields a $21 \times 20 = 420$ dimensional latent space. See below for more details. We observed that various aspects of model performance such as reconstruction loss and DCI scores increased with depth, corroborating previous work [71]. We set the initial number of channels to 32 in the first layer of the encoder and doubled the channel size every time we downsample the features spatially. This resulted in a width of 256 in the final layer. Please refer to our code for more information.

### 9.3 cNVAE vs. NVAE latent dimensionality

As an illustrative example, here we count the number of latent variables in cNVAE and compare them to an otherwise identical but non-compressed NVAE. The total number of latent variables for a hierarchical VAE is given by:

$$D = \sum_{\ell=1}^{L} d_\ell, \tag{14}$$

Table 4: The hierarchical models (cNVAE, cNAE) process information at various spatial scales. These models have a total of 21 latent groups, each containing $d = 20$ latent variables. For hierarchical models, we concatenate the latent representations across all levels of the hierarchy, resulting in a single latent vector with dimensionality $D = 420$. In contrast, non-hierarchical models (VAE, AE) do not possess such a multi-scale organization. Instead, their non-hierarchical latent spaces contain a single latent group, operating at a single spatial scale. This single latent group contains the entire set of $d = D = 420$ latent variables. See Table 2 and Fig. 2.

| Model | Number of latent groups | | | | Number of latent variables per group | Latent space dimensionality |
|-------|-------|-------|-------|-------|-------|-------|
| | $2 \times 2$ | $4 \times 4$ | $8 \times 8$ | total | | |
| cNVAE | 3 | 6 | 12 | 21 | $d = 20$ | $D = 21 \times 20 = 420$ |
| VAE | 1 | – | – | 1 | $d = 420$ | $D = 1 \times 420 = 420$ |
| cNAE | 3 | 6 | 12 | 21 | $d = 20$ | $D = 21 \times 20 = 420$ |
| AE | 1 | – | – | 1 | $d = 420$ | $D = 1 \times 420 = 420$ |

where $d_\ell$ is the number of latents in level $\ell$, and $L$ is the total number of hierarchical levels. As described above, $d_\ell = d \cdot s_\ell^2$ for NVAE, where $d$ is the number of latents per level and $s_\ell$ is the spatial scale of level $\ell$. In contrast, this number is a constant for the cNVAE: $d_\ell = d$. In this paper, we set $d = 20$ and had the following configuration for the cNVAE:

- "top" latents: 3 groups operating at the spatial scale of $2 \times 2$
- "mid" latents: 6 groups operating at the spatial scale of $4 \times 4$
- "bottom" latents: 12 groups operating at the spatial scale of $8 \times 8$

This resulted in an overall latent dimensionality of $D = (3 + 6 + 12) \times d = 21 \times 20 = 420$ for the cNVAE. In contrast, an NVAE with a similar number of hierarchical levels would have a dimensionality of $D = (3 \cdot 2^2 + 6 \cdot 4^2 + 12 \cdot 8^2) \times d = 876 \times 20 = 17,520$.

Note that in Fig. 4, we necessarily had to reduce the NVAE depth to roughly match its latent dimensionality to other models. Specifically, the NVAE in Fig. 4 had $2/3/6$ latent groups at top/mid/bottom with $d = 1$. This configuration resulted in latent dimensionality of $D = (2 \cdot 2^2 + 3 \cdot 4^2 + 6 \cdot 8^2) \times 1 = 440$, which is roughly comparable to other models at $D = 420$. See Table 4 for a comparison between cNVAE and the alternative models used in this study.

## 9.4  Model training details

We generated $750,000$ samples for each ROFL category, with a $600,000/75,000/75,000$ split for train/validation/test datasets. Model performance (reconstruction loss, NELBO) was evaluated using the validation set (Fig. 14 and 15). Similarly, we used the validation set to estimate mutual information between latents, $\boldsymbol{z}$, and ground truth factors, $\boldsymbol{g}$, as shown in Fig. 3. To evaluate the latent codes, we used the validation set to train linear regressors to predict $\boldsymbol{g}$ from $\boldsymbol{z}$. We then test the performance using the test set. This includes results presented in Fig. 4 and Fig. 5.

**KL annealing and balancing.**  We annealed the KL term during the first half of the training, which is known to be an effective trick in training VAEs [52, 72, 132, 133]. Additionally, during the annealing period, we employed a KL balancing mechanism which ensures an equal amount of information is encoded in each hierarchical latent group [52, 134, 135].

**Gradient norm clipping.**  During training, we occasionally encountered updates with large gradient norms. Similar to vdvae [71], we used gradient clipping, with an empirically determined clip value of 250. However, we did not skip the updates with large gradients, as we found that the model usually gets stuck and does not recover after skipping an update.

**Training hyperparameters.**  We used batch size of 600, learning rate of 0.002, and trained models for 160 epochs (equivalent to $160k$ steps). Each training session took roughly 24 hours to conclude on a single Quadro RTX 5000 GPU. Similar to NVAE, we used the AdaMax optimizer [136] with a cosine learning rate schedule [137] but without warm restarts.

## 9.5 Choosing $\beta$ values for different figures

It is known that different choices of the $\beta$ value in the VAE loss (Table 2) will result in different model properties [85]. Therefore, while we scanned across many $\beta$ values spanning two orders of magnitude, we displayed results for certain betas to best demonstrate our points.

To ensure a fair model comparison in the results shown in Figs. 3, 4, 11 and 12, we selected $\beta$ values that maximized the overall untangling score for each architecture (cNVAE, $\beta = 0.15$; VAE, $\beta = 1.5$). This can be seen in Fig. 5, which presents DCI scores over the entire $\beta$ spectrum. The top row of Fig. 5 displays informativeness (= untangling) scores, revealing an inverted U shape, consistent with previous work [85]. Different architectures peak at different $\beta$ values.

For the neuron shown in Fig. 8b, we found that different $\beta$ values lead to qualitatively similar outcomes, that is, extreme sparsity in cNVAE feature importances, in sharp contrast to that of VAE. We deliberately chose to show a larger $\beta = 5$ for VAE because previous work by Higgins et al. [16] suggested that increasing $\beta$ values increases neuronal alignment, which we did not observe here. In contrast, even a very small $\beta = 0.01$ for the cNVAE results in a large alignment. This result (paired with other observations) suggests that brain alignment, as we measured it here, emerges due to the architecture rather than from large $\beta$ values alone—although we do observe some dependence on $\beta$, so ultimately a combination of both architecture and loss is important (but mostly architecture).

## 9.6 Details on latent code evaluation

Each ROFL sample $x \in \mathbb{R}^{2 \times N \times N}$ is rendered from a low dimensional ground truth vector $g \in \mathbb{R}^K$, where $N$ is the number of pixels (stimulus size) and $K$ is the true underlying dimensionality of the data (i.e., number of independent ground truth factors). In other words, the data lies on a curved $K$-dimensional manifold embedded in a $2N^2$-dimensional space.

In the present work, we worked with relatively small stimuli with $N = 17$. This choice allowed us to train more models and explore a larger combination of hyperparameters. We chose an odd value for $N$ so that there was a pixel at the exact center, which served as the center of gaze or the fixation point (Fig. 1a). The true dimensionality (number of generative factors) of different ROFL categories is given in Table 1. For example, it is $K = 11$ for `fixate-1`.

We trained unsupervised models by feeding them raw samples $x$ and asking them to produce a reconstruction $\hat{x}$. The models accomplish this by mapping their inputs to latent codes $z$ during inference (red pathways in Fig. 2), and mapping $z$ to $\hat{x}$ during generation (blue pathways in Fig. 2). Note that for all models we have $z \in \mathbb{R}^{420}$ (see Table 4).

In sum, our approach consists of four steps:

1. Data generation: $g \rightarrow \boxed{\text{ROFL}} \rightarrow x$
2. Unsupervised training: $x \rightarrow \boxed{\text{Model}} \rightarrow \hat{x}$
3. Inference: $x \rightarrow \boxed{\text{Model}} \rightarrow z$
4. Evaluation: $z \leftrightarrow g$ ?

Within this framework, we investigated the relationships between data generative factors $g$ and model-learned latent codes $z$ by using the DCI metrics, which we discuss in detail below.

## 9.7 Metrics & Baselines

We trained a variety of unsupervised models (Table 2) and evaluated them based on (1) their ability to represent the generative variables of the ROFL stimuli; and (2) the neural alignment of their representation. This resulted in a total of five different metrics used to evaluate model performance. Below, we describe the different metrics used for each.

**#1: untangling & disentangling of data generative factors.** To evaluate the latent codes, we employed the DCI framework [22] and made use of the fact that ROFL provides ground truth factors for each data sample. DCI is comprised of three different metrics, *Disentanglement*, *Completeness*, and *Informativeness*, which we will discuss in detail below.

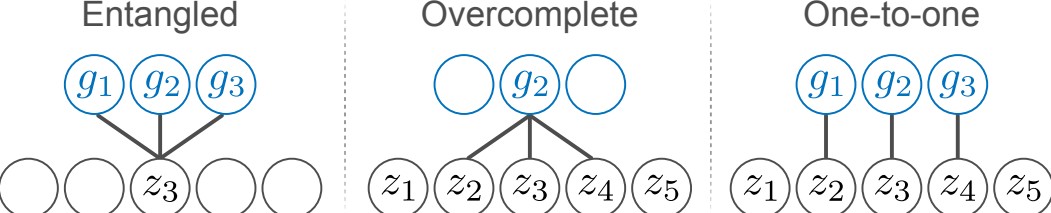

Figure 9: Suppose we train a regression model to predict ground truth factors $g_j$ (blue) from latents $z_i$ (grey). Entanglement (left) refers to a situation in which one latent variable contains information about many generative factors. An overcomplete code (middle) is one in which a single generative factor is predicted from many latent variables. A one-to-one relationship (right) is the ideal outcome.

**#2: explaining variability in biological neurons & brain alignment.** We used ridge regression to predict macaque MT neuron responses from model latents. To measure the alignment between different latent codes and biological neurons, we used two standard metrics: linear predictive power and brain-alignment score defined in Fig. 8a.

### 9.7.1 The DCI framework

In this section, we provide more details about the DCI framework [22] and discuss how it was used to evaluate the disentangling performance of our models. The core motivation behind DCI is that a good latent code ideally disentangles as many factors of variation as possible while discarding as little information as possible. In other words, we consider a latent code to be "*good*" as long as:

1. It can predict the variance of data generative factors, and
2. It exhibits a one-to-one relationship with the generative factors (Fig. 9).

To make this idea concrete, the starting point is to train a regressor $f_j$ (e.g., lasso) that predicts a given data generative factor $g_j$ from a latent code $z$. The performance of $f$ in predicting $g$ (measured using e.g., $R^2$) is referred to as the *Informativeness* score. The notion of *Informativeness*, if measured using a linear regressor, is very similar to *Untangling* [20].

As the next step, the weights of $f$ are used to construct a matrix of relative importances $R$, where $R_{ij}$ denotes the relative importance of $z_i$ in predicting $g_j$. For example, $R$ can be the magnitude of lasso coefficients. The matrix of relative importance is then used to define two pseudo-probability distributions:

$$P_{ij} = \frac{R_{ij}}{\sum_k R_{ik}} = \text{probability of } z_i \text{ being important for predicting } g_j \qquad (15)$$

$$\tilde{P}_{ij} = \frac{R_{ij}}{\sum_k R_{kj}} = \text{probability of } g_j \text{ being predicted by } z_i \qquad (16)$$

These distributions are then used to define the following metrics, which complete the DCI trinity:

$$Disentanglement: \quad D_i = 1 - H_K(P_{i.}); \quad \text{where} \quad H_K(P_{i.}) = -\sum_j P_{ij} \log_K P_{ij} \qquad (17)$$

$$Completeness: \quad C_j = 1 - H_D(\tilde{P}_{.j}); \quad \text{where} \quad H_D(\tilde{P}_{.j}) = -\sum_i \tilde{P}_{ij} \log_D \tilde{P}_{ij} \qquad (18)$$

Here, $K$ is the number of data generative factors (e.g., $K = 11$ for `fixate-1`, Table 1), and $D$ is the number of latent variables ($D = 420$ for the models considered in our paper, Table 4).

Intuitively, if latent variable $z_i$ contributes to predicting a single ground truth factor, then $D_i = 1$. Conversely, $D_i = 0$ when $z_i$ equally contributes to the prediction of all ground truth factors. If the ground truth factor $g_j$ is being predicted from a single latent variable, then $C_j = 1$ (a complete code).

On the other hand, if all latents contribute to predicting $g_j$, then $C_j = 0$ (maximally overcomplete). Finally, the overall disentanglement or completeness scores are obtained by averaging across all latents or generative factors.

In the DCI paper, Eastwood and Williams [22] used lasso or random forest as their choice of regressors $f$. Initially, we experimented with lasso but found that the lasso coefficient had a strong impact on the resulting DCI scores. To mitigate this, we used linear regression instead and estimated the matrix of relative importance using scikit-learns's permutation importance score (sklearn.inspection.permutation_importance), which measures how much performance drops based on the shuffling of a given feature.

### 9.7.2 Brain alignment score

As depicted in Fig. 8a, the alignment score emphasizes the degree of correspondence between latent variables and neurons. A sparse relationship between latents and neurons suggests a higher alignment, indicating that the model's representation closely mirrors the neuron's representational "form" [16]. This notion of alignment is intimately tied to the "completeness" score discussed above. See Fig. 10.

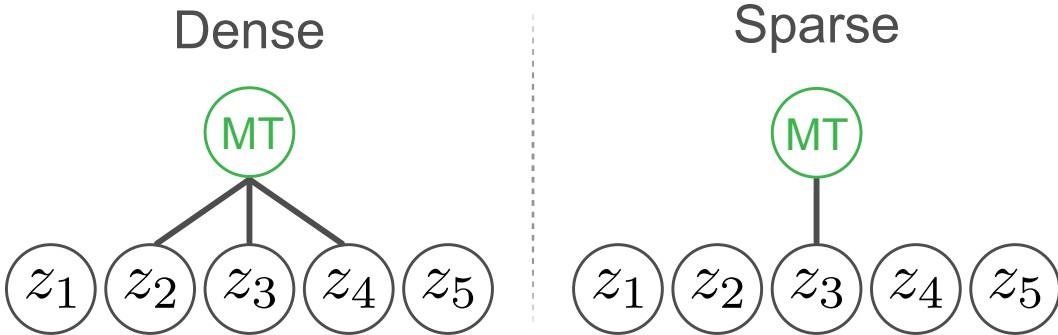

Figure 10: Consider training a regressor to predict MT neuron responses (green) from latent variables $z_i$ (grey). Beyond accurate predictions, we assess the alignment between $z$ and MT based on the sparsity of the resulting latent-to-neuron relationships. One possibility is that many latents contribute to predicting the neuron in a dense fashion. This would suggest that the neuron's function is diffusely represented within the latents, hinting that the representational form of the latents is ultimately different from that of MT, perhaps through rotation or affine transformations. Conversely, when few latents effectively predict MT, it suggests a similar representational form, laying the foundation for our rationale in defining our alignment score (Fig. 8a).

### 9.7.3 Baselines

In Fig. 4 we compared the cNVAE with alternative models introduced in Table 2. We also included an NVAE with matched number of latents, but necessarily fewer hierarchical levels (see section 9.3 for more details). To demonstrate the difficulty of our untangling task, we included two simple baselines: Raw data and a linear code based on the first 420 dimensions of data principal components (PCA). Both Raw and PCA baselines performed poorly, only weakly untangling self-motion velocity ($\sim 55\%$) and object velocity in $x$ and $y$ directions ($47\%$). This result demonstrates the importance of nonlinear computation in extracting good representations.

## 10 Additional Results

In this section, we present additional results demonstrating the advantages of hierarchical VAEs versus non-hierarchical ones, which support the results in the main text. All models explored in this section were trained on fixate-1. For both cNVAE and VAE, we chose $\beta$ values that maximized their informativeness (we discuss this choice in section 9.5). Specifically, we set $\beta = 0.15$ for cNVAE and $\beta = 1.5$ for VAE. These same models trained using the respective $\beta$ values were also used for results presented in Fig. 3 and Fig. 4 in the main text.

## 10.1 Individual cNVAE latents exhibit greater linear correlation with ground truth factors

In the main text (Fig. 4) we demonstrated that the cNVAE latent representations $z$ contained explicit information about ground truth factors $g$. We achieved this by training linear regression models that map the $420$-dimensional latent space to each ground truth factor. Here, we investigate the degree that *individual* latents $z_i$ are (Pearson) correlated with individual factors $g_j$ (Fig. 11). For some ground truth factors, we find that cNVAE learns latents that are almost perfectly correlated with them. For instance, $r = -0.91$ for $F_x$ and $r = 0.87$ for $F_y$, compared to modest correlation values of $r = 0.34$ and $r = 0.38$ for VAE. The higher linear correlation is consistently observed for cNVAE over VAE. Especially, we note that the VAE does not capture object-related variables at all, consistent with other results in the main paper (Figs. 3 and 4).

For the cNVAE, the indices for the selected highly correlated latents are as follows: $[296, 248, 393, 284, 368, 92, 206, 105, 338, 60, 63]$, ordered left-to-right in the same manner as in Fig. 11. Here, an index of $0$ corresponds to the very top latent operating at the spatial scale of $2 \times 2$, and an index of $419$ corresponds to the very bottom one at scale $8 \times 8$ (see Fig. 2). By considering the hierarchical grouping of these latents (see dashed lines in Fig. 3) we see that the majority of the highly correlated latents operate at the spatial scale of $8$, with the exception of object-related ones corresponding to $X_{obj}$, $Z_{obj}$, $V_{obj,y}$, and $V_{obj,z}$ which operate at $4 \times 4$.

In conclusion, Fig. 11 revealed that cNVAE learns individual latents that are highly correlated with individual ground truth factors, which provides complementary insights as to why cNVAE is so successful in untangling (Fig. 4). However, we note that a high correlation with individual latents is not the best way to capture the overall functional organization, which is better reflected for example in the mutual information heatmap of Fig. 3.

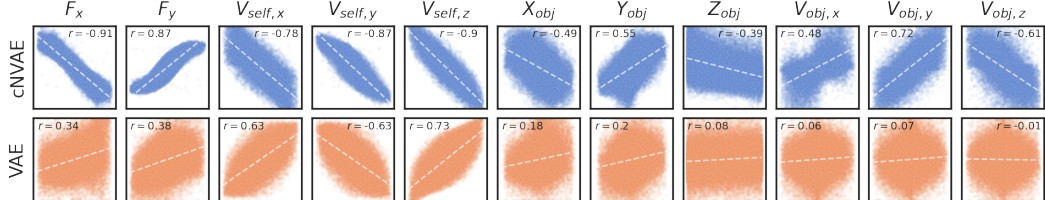

Figure 11: For each ground truth factor and model, we identified a single latent variable that displayed the highest (Pearson) correlation with that variable. The ground truth factors exhibited greater linearity within the latent space of cNVAE compared to VAE. Scatter points correspond to data samples from the test set. x-axis, the value of ground truth factors; y-axis, the value of latent variables.

## 10.2 Generated samples: non-hierarchical VAE completely neglects objects

If a generative model has truly learned the underlying structure of a dataset, then it should be able to generate fake samples that look like its training data. Here, we examine generated samples from both the cNVAE and VAE models, as illustrated in Fig. 12. For comparison, we also include five samples randomly drawn from the `fixate-1` dataset that these models were trained on (Fig. 12, first row). The cNVAE samples appear indistinguishable from true samples; however, VAE fails to capture and generate objects, corroborating our previous results (see Fig. 3, Fig. 4, and Fig. 11). This striking divergence between cNVAE and VAE indicates that a hierarchically organized latent space is necessary for effectively modeling datasets that contain multiple interacting spatial scales.

## 10.3 Latent traversal reveals multi-scale understanding in cNVAE

One desired property of VAE latent representations is that continuous interpolation along a latent dimension results in interpretable effects on the corresponding reconstruction. Here, we provide visual examples of latent traversal in cNVAE, focusing on a latent dimension that effectively captures $Y_{obj}$ (Figure 13). These examples demonstrate intriguing interactions between object velocity patterns and self-motion within the scene. Notably, in some examples such as rows 0 and 6, relocating the object from the bottom to the top of the scene induces a significant change in its linear velocity along the y-axis. This phenomenon arises due to the influence of self-motion, as seen in the background.

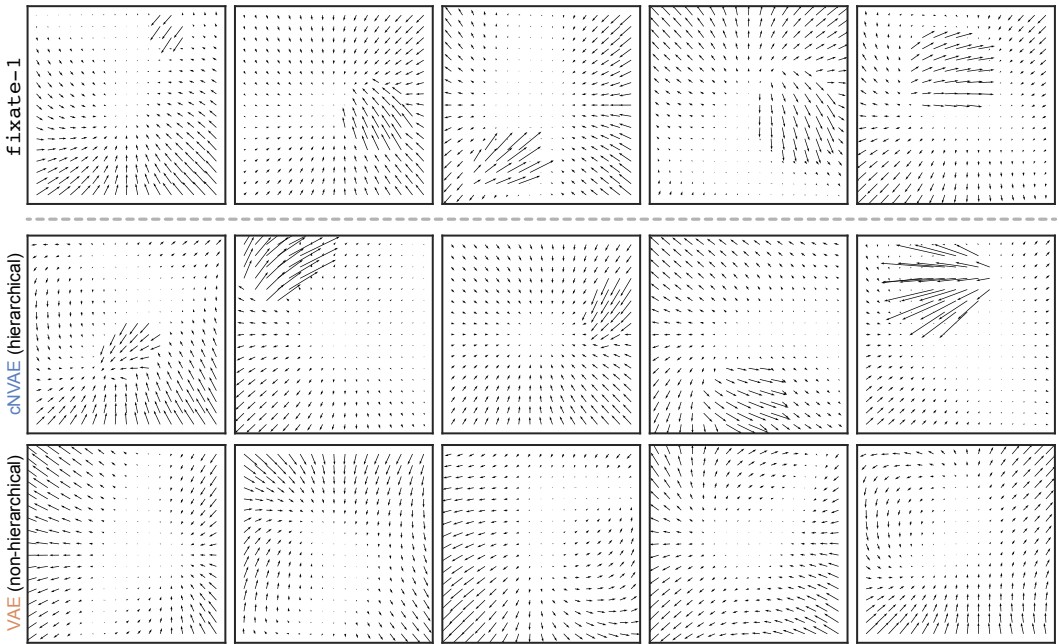

Figure 12: The first row shows random samples drawn from ROFL `fixate-1` category. The second and third rows are samples generated from cNVAE and VAE models that were trained on `fixate-1`. Generated samples from cNVAE closely resemble real data; whereas, VAE ignores objects. Samples were generated at a temperature of $t = 1$ without cherry-picking.

These illustrations collectively showcase that flow vectors in the visual scene are shaped both by local variables, such as the object's physical location, and their interactions with global-influencing variables like self-motion. By comprehending data across scales, cNVAE is able to capture such hierarchical and multi-scale properties inherent in naturalistic optic flow patterns. In contrast, non-hierarchical VAEs exhibit a complete failure in this regard.

### 10.4 All models achieve good reconstruction performance

All models considered in this paper are trained based (in part or fully) on reconstruction loss, corresponding to how well they could reproduce a presented stimulus. Here we report the models' reconstruction performance, to ensure that they are all able to achieve a reasonable performance. Figure 14 illustrates how the reconstruction loss (EPEPD, endpoint error per dim) and negative evidence lower bound (NELBO) depend on different values of $\beta$ and model types. Both hierarchical and non-hierarchical VAE models perform well, with cNVAE exhibiting slightly superior performance for $\beta < 0.8$.

### 10.5 Untangling and reconstruction loss are strongly anti-correlated for cNVAE but not VAE

In this work, we demonstrated that a virtue of hierarchical VAEs is that they make information about ground truth factors easily (linearly) accessible in their latent representations. We were able to quantify this relationship because we used a simulation (ROFL) where ground truth factors were available. However, such a condition is not necessarily the case in real-world datasets that typically lack such direct knowledge [22].

To address this, we examined the relationship between informativeness and reconstruction loss, as well as NELBO, depicted in Fig. 15, to test whether there is a relationship between them. Strikingly, we observe a significant and strong negative linear relationship between reconstruction loss and informativeness in cNVAE ($r = -0.91, p = 8.9 \times 10^{-7}, t-$test). Conversely, no such correlation is observed for VAE ($r = 0.10, p = 0.72, t-$test). Notably, this relationship is absent for both models when comparing informativeness with NELBO (Fig. 15). These findings suggest that reconstruction

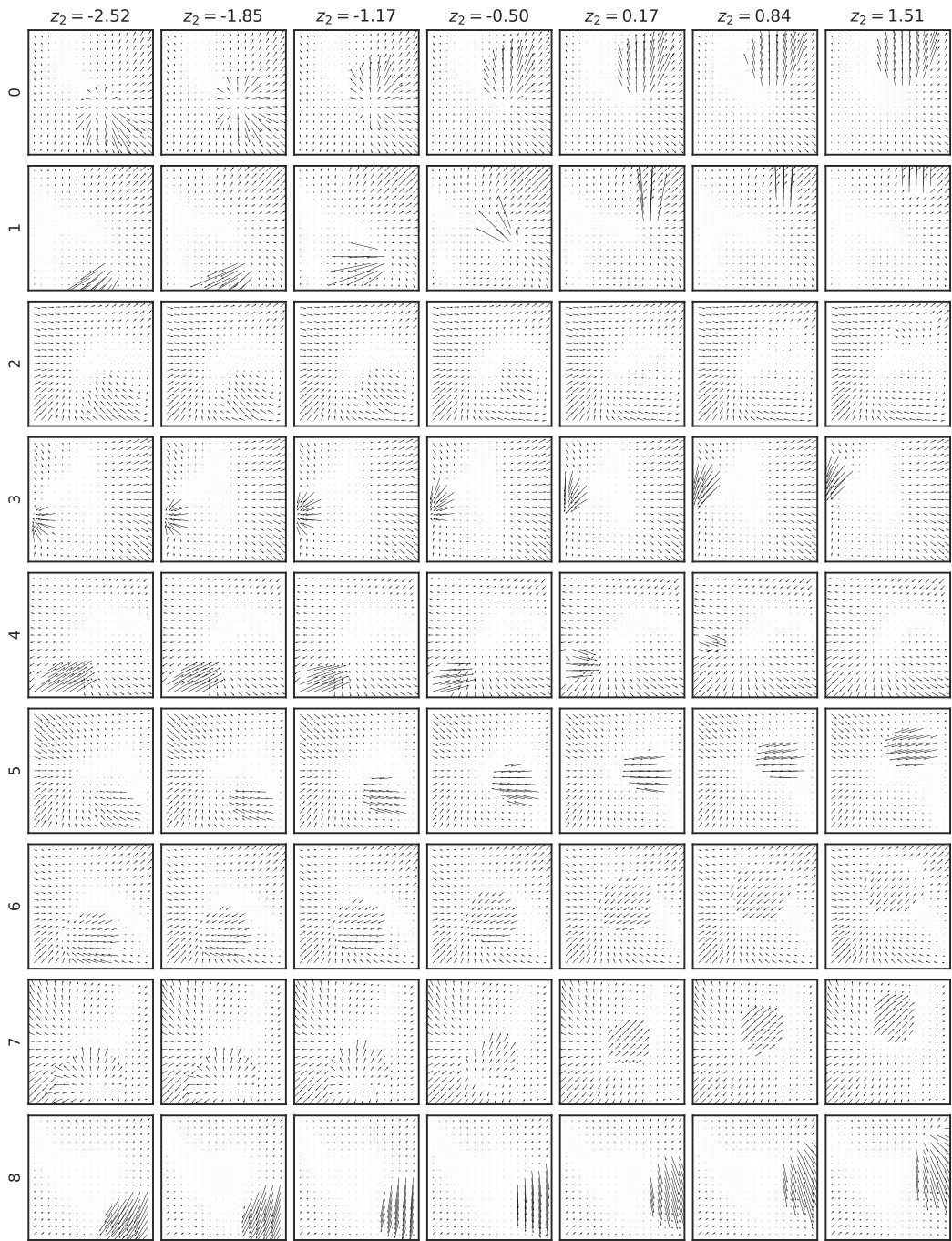

Figure 13: Latent traversal performed using a latent variable that effectively captures $Y_{obj}$. This variable belongs to a latent group operating at the spatial scale of $2 \times 2$ (see Figs. 2 and 3; table 4).

loss—exclusively in the case of cNVAE—can serve as a viable proxy for informativeness, even in the absence of ground truth factors in real-world datasets.

## 10.6 Brain alignment requires more than just linear regression performance

In the field of neuroscience, linear regression $R^2$ scores have commonly been employed to measure alignment between biological and artificial neural networks [30, 31, 34, 36]. However, we found

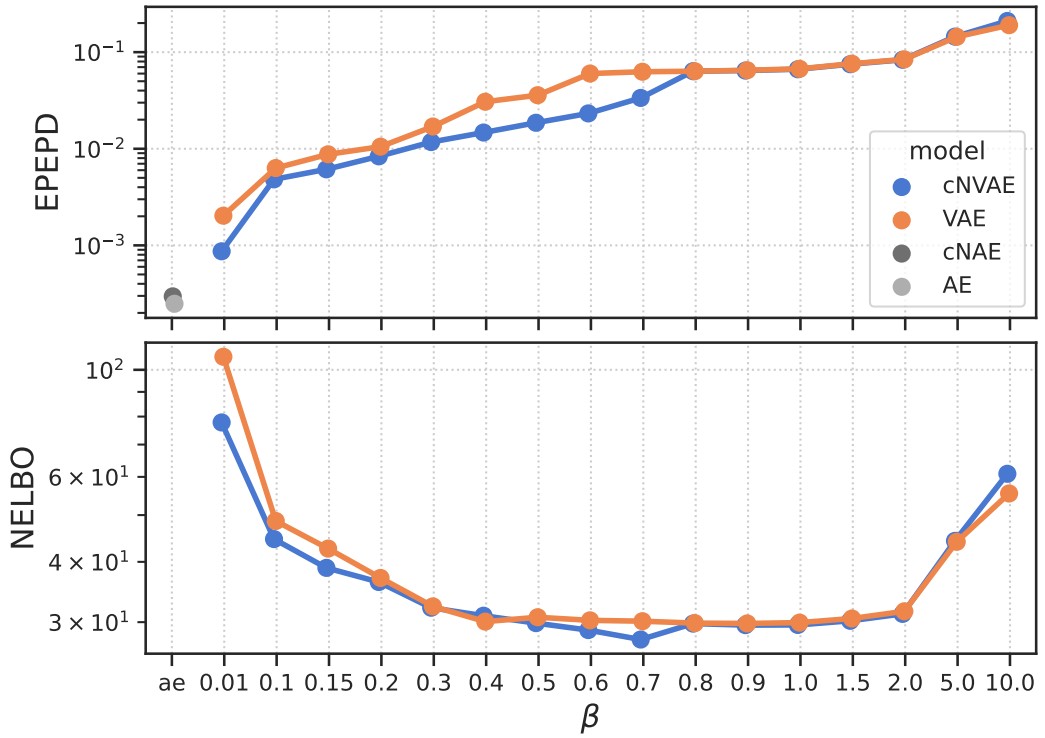

Figure 14: Both models demonstrate robust reconstruction performance, with cNVAE exhibiting a slight improvement. EPEPD, endpoint error per dim (lower is better). NELBO, negative evidence lower bound (lower is better).

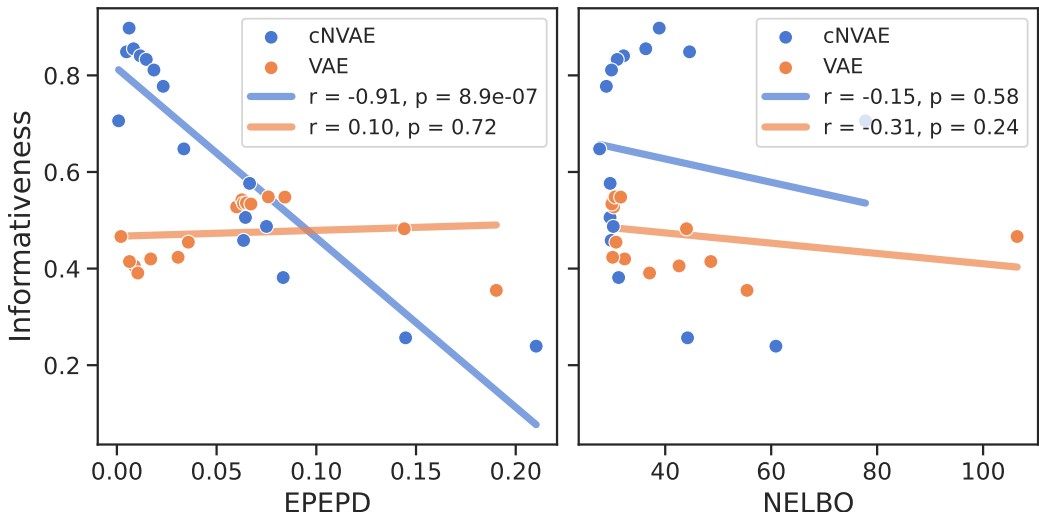

Figure 15: Strong anti-correlation between reconstruction loss and informativeness (i.e., untangling) observed only for cNVAE. Each scatter point corresponds to a model trained with a different $\beta$ value. EPEPD, endpoint error per dim (lower is better). NELBO, negative evidence lower bound (lower is better).

that relying solely on regression performance provided a weak signal for model selection, which is consistent with recent work [36–38, 112, 125]. In contrast, our alignment score defined in Fig. 8a provided a much stronger signal for differentiating between models (compare Fig. 7 and Fig. 8c).

To investigate this further, we conducted statistical tests, comparing pairs of models based on either their performance in explaining the variance of MT neurons or their brain-alignment scores. The results are presented in Table 5. We observed that ridge regression performance was not particularly effective in distinguishing between models. We measured the effect sizes between distributions of $N = 141$ neurons, measuring how well ridge regression performance (Fig. 7) or alignment scores (Fig. 8c) discriminated between models. We found that the effect sizes were substantially larger for the alignment scores (Fig. 16). These findings suggest that while model performance is necessary, it is not sufficient for accurately measuring alignment between artificial and biological representations. In conclusion, refined and more comprehensive approaches are required to properly capture brain alignment [112–115].

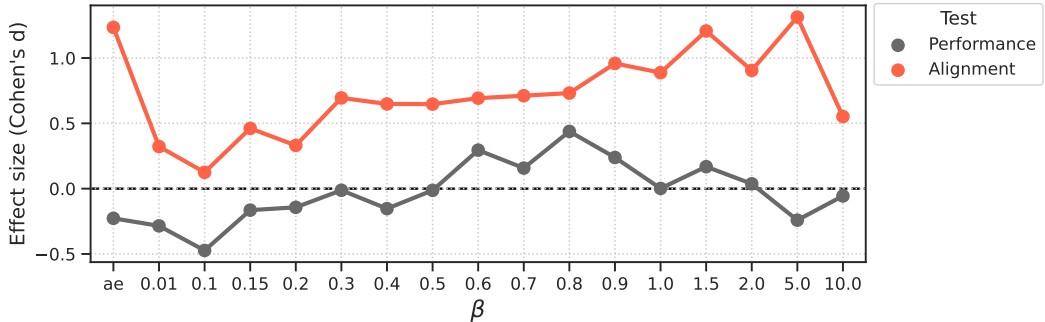

Figure 16: Effect sizes are shown for the two different approaches in model comparisons: using ridge regression *Performance* (related to Fig. 7), versus, *Alignment* scores (related to Fig. 8c). Alignment demonstrates a substantially larger effect size, leading to a more pronounced differentiation among the models. Positive values indicate a larger value for hierarchical models (i.e., cNVAE > VAE or cNAE > AE). For different $\beta$ values, model *Performance* difference fluctuates between positive and negative values. In contrast, the difference in model *Alignment* scores is consistently positive, indicating a greater brain alignment for hierarchical models.

## 10.7 Dissecting cross-validated model performance on neural data

One measure of neural alignment that we consider here used unsupervised latents to predict experimentally recorded neural responses (via linear regression). The neurons in this dataset had long presentations of optic flow stimuli, but only a fraction of the neurons (105 out of 141 neurons) had multiple repeats of a short stimulus sequence. Because these repeats facilitated a much clearer measure of model performance, we used these where available, following previous work [34]. However, due to the short stimulus sequence ($\sim 5$ sec; see Fig. 6b) used to estimate performance in the neurons with stimulus repeats, here we perform additional tests to verify that our conclusions hold using a more rigorous cross-validation procedure.

Specifically, here we used the long stimulus sequence to both train model parameters (80% of data) and determine hyperparameters (including regularization and optimal latencies) using a separate validation set (20%). We then predicted model performance on the repeats. This is in contrast to our approach in the main text, which used the repeat data itself to optimize hyperparameters, but as a result, risked overfitting on the short stimulus sequence. Indeed, we found that independent determination of hyperparameters (from the validation set) resulted in decreased model performance on the test set (Table 6) for both cNVAE and VAE models, although with comparable effect sizes. As a result, our conclusions of overlapping performance between cNVAE and VAE with a slightly better performance of the cNVAE were robust.

| $\beta$ | Performance (Figure 7) | | | Alignment scores (Figure 8c) | | |
|---|---|---|---|---|---|---|
| | effect size | $p-$value | significant? | effect size | $p-$value | significant? |
| ae | -0.23 | 0.01 | ✓ | 1.2 | 8e-29 | ✓ |
| 0.01 | -0.28 | 0.002 | ✓ | 0.32 | 0.0004 | ✓ |
| 0.1 | -0.47 | 2e-07 | ✓ | 0.12 | 0.2 | ✗ |
| 0.15 | -0.16 | 0.07 | ✗ | 0.46 | 4e-07 | ✓ |
| 0.2 | -0.14 | 0.1 | ✗ | 0.33 | 0.0003 | ✓ |
| 0.3 | -0.012 | 0.9 | ✗ | 0.69 | 4e-13 | ✓ |
| 0.4 | -0.15 | 0.09 | ✗ | 0.65 | 7e-12 | ✓ |
| 0.5 | -0.013 | 0.9 | ✗ | 0.65 | 7e-12 | ✓ |
| 0.6 | 0.29 | 0.001 | ✓ | 0.69 | 4e-13 | ✓ |
| 0.7 | 0.16 | 0.08 | ✗ | 0.71 | 1e-13 | ✓ |
| 0.8 | 0.44 | 1e-06 | ✓ | 0.73 | 4e-14 | ✓ |
| 0.9 | 0.24 | 0.008 | ✓ | 0.96 | 1e-20 | ✓ |
| 1.0 | 0.0016 | 1 | ✗ | 0.89 | 1e-18 | ✓ |
| 1.5 | 0.17 | 0.07 | ✗ | 1.2 | 3e-28 | ✓ |
| 2.0 | 0.037 | 0.7 | ✗ | 0.91 | 3e-19 | ✓ |
| 5.0 | -0.24 | 0.008 | ✓ | 1.3 | 7e-31 | ✓ |
| 10.0 | -0.056 | 0.6 | ✗ | 0.55 | 3e-09 | ✓ |

Table 5: Hierarchical architectures are significantly more aligned with MT neurons compared to non-hierarchical ones. Paired $t-$tests were performed, and $p-$values were adjusted for multiple comparisons using the Benjamini-Hochberg correction method [138]. Effect sizes were estimated using Cohen's $d$ [139], where positive values indicate cNVAE > VAE (or cNAE > AE, first row).

| Model | Performance, $R$ $(\mu \pm se;\ N = 105)$ | | | |
|---|---|---|---|---|
| | $\beta = 0.5$ | $\beta = 0.8$ | $\beta = 1$ | $\beta = 5$ |
| cNVAE | $.452 \pm .025$ | $.479 \pm .025$ | $.486 \pm .024$ | $.462 \pm .025$ |
| VAE | $.461 \pm .027$ | $.401 \pm .029$ | $.466 \pm .028$ | $.461 \pm .027$ |
| cNAE | | | $.460 \pm .026$ | |
| AE | | | $.412 \pm .031$ | |

Table 6: Model performance on the subset of neurons for which we had repeat data ($N = 105$). Here, we used the repeat data as a held-out test set. We found that model performances remain statistically indistinguishable, providing additional support for our main results.

# 11 Additional Discussion

## 11.1 Visual-only cues and unsupervised learning are sufficient for untangling optic flow causes

It is well known that an observer's self-motion through an environment can be estimated using optic flow patterns alone [80], although it has been proposed that the brain also uses a combination of visual and non-visual cues [140], such as efference copies of motor commands [141], vestibular inputs

[142], and proprioception [143]. Indeed, using optic flow alone can be complicated by rotational eye movements and independently moving objects [48, 144, 145]. How can the brain estimate the motion of an object when self-motion is non-trivially affecting the entire flow field [146, 147], including that of the object itself? This is a critical motion processing problem [148], manifested in diverse ecological scenarios such as prey-predator dynamics, crossing the street, or playing soccer. Are retinal visual-only cues sufficient to extract information about different self-motion components, as well as object motion [149]?

Unlike the approach of Mineault et al. [34], our work is based on the premise that retinal-only signals contain sufficient information about their ground truth causes and that these factors can be extracted using unsupervised learning alone. In contrast, Mineault et al. [34] trained their DorsalNet by providing dense supervision on self-motion components, arguing that such supervised training is biologically plausible since these signals are approximately available to the observer from vestibular inputs and other such extra-retinal sources.

Our results (Fig. 4) provide proof-of-concept that it is possible to untangle ground truth causes of retinal optic flow in a completely unsupervised manner, using the raw optic flow patterns alone. Comparing the performance of alternative models revealed two essential ingredients for the success of our approach: (1) an appropriate loss function—inference [1]; and, (2) a properly designed architecture—hierarchical, like the sensory cortex [3].

## 11.2 Comparing cNVAE to NVAE, its non-compressed counterpart

In Fig. 4, we investigated the performance of the original non-compressed NVAE with an approximately matched latent dimensionality (440, slightly larger than 420 for cNVAE: more details are provided in section 9.3). We found that although NVAE did better than non-hierarchical VAE, it still underperformed cNVAE. This is because trying to match the latent dimensionality of NVAE with cNVAE necessitates reducing the number of hierarchical latent groups in the NVAE since most of its convolutional latents capture spatial dimensions rather than more abstract features as cNVAE does. From previous work [71], we know that the "stochastic depth" of hierarchical VAEs is the key reason behind their effectiveness; therefore, it was expected that reduced depth would hurt an NVAE with a matched number of latents.

It is worth noting that cNVAE and NVAE had a balanced performance across all ground truth factors (Fig. 4), in that they captured both global (self-motion-related) and local (object-related) aspects well. In contrast, non-hierarchical VAE completely ignored object-related variables and focused solely on the global scale determined by fixation point and self-motion. This suggests that hierarchical architecture is crucial for a more comprehensive understanding of multi-scale datasets such as ROFL.

## 11.3 Disentanglement is in the eyes of the beholder

The conventional definition of disentanglement in machine learning relies on relating the latent representation to predefined generative factors, a choice that is often not uniquely determined, and is often guided by heuristics [18]. For example, in the present work, we chose to represent velocities in Cartesian coordinates, rather than polar coordinates and judged the degree of disentanglement based on the relationship of the latents to these factors. Notably, these coordinate systems are related by a nonlinear transform, and latents linearly related to one coordinate system will then be less linearly related to the other.

Throughout our experiments, we observed one or more cNVAE latent variables that exhibited equally high mutual information with all components of self-motion velocity in Cartesian coordinates (e.g., latent dimensions 154 and 122 in Fig. 3). When we translated the generative factors into polar coordinates and looked at their relationship with these latents, we found that these latents were solely encoding the magnitude of self-motion, irrespective of its direction (Fig. 17). Thus, these seemingly highly entangled latent variables would have achieved almost perfect disentanglement scores had we employed polar coordinates instead of Cartesian coordinates to represent the ground truth velocities ($d_i \approx 0.38 \rightarrow d_i \approx 0.81$).

Our findings shed light on the fact that disentanglement cannot be considered an inherent property of objective reality; rather, it is significantly influenced by our a priori determination of independent factors of variation within the data. Given the inherent complexity and richness of natural datasets, it

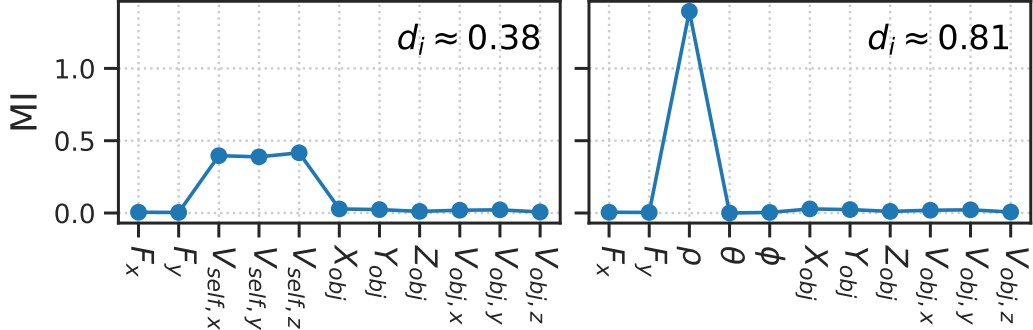

Figure 17: This latent variable cares about the magnitude of self-motion only. Its disentanglement score depends strongly on what we choose a priori as our independent factors of variation in data.

becomes challenging to unequivocally determine the true factors of variation a priori, pointing to the need for a more systematic approach.

When it comes to interpreting brain functions, perhaps considering the behavioral relevance of generative factors could guide our choices. For example, an animal directing its own movements does not pick grid coordinates but rather chooses direction and speed, possibly making a polar representation more "behaviorally relevant" to representations within its brain.

Another promising approach lies in focusing on the symmetry transformations inherent in the world [18, 150]. By seeking representations that respect the underlying geometrical and group-theoretical structures of the world [151, 152], and comparing them in principled ways [112–114], we may uncover new perspectives that provide valuable insights beyond the conventional and subjective notions of disentanglement.

## 12    Additional Background & Related Work

The deep connections between VAEs and neuroscience is explored in a review by Marino [17], but work looking at the synergy between neuroscience and VAEs has thus far been limited to the following recent papers.

Storrs et al. [15] trained PixelVAE [153] models on synthesized images of glossy surfaces and found that the model spontaneously learned to cluster images according to underlying physical factors like material reflectance and illumination, despite receiving no explicit supervision. Most notably, PixelVAE mimicked human perceptual judgment of gloss in that it made errors that were similar to those made by humans.

Higgins et al. [16] trained $\beta$-VAE [85] models on face images and evaluated the models on predicting responses of individual inferotemporal (IT) neurons from Macaque face patch. They found that $\beta$-VAE discovers individual latents that are aligned with IT neurons and that this alignment linearly increases with increased beta values.

Csikor et al. [68] investigated the properties of representations and inference in a biologically inspired hierarchical VAE called Topdown VAE that is capable of reproducing some key properties of representations emerging in V1 and V2 of the visual cortex. The authors showed that architectures that feature top-down influences in their recognition model give rise to a richer representation, such that specific information that is not present in mean activations becomes linearly decodable from posterior correlations.

Finally, Miliotou et al. [69] learned mapping from the latent space of a hierarchical VAE to fMRI voxel space. This allowed the use of the trained decoder to reconstruct images directly from brain responses. Importantly, they perform ablation experiments and find hierarchy is an essential component of their models.

# 13 Appendix: ROFL derivations and detailed description

ROFL aims to generate synthetic optical flow patterns that closely resemble those arising from ecologically relevant types of motion. This entails the consideration of both self-motion and object-motion components. Specifically, the self-motion component encompasses various subtypes such as translational and rotational motions induced by eye or head movements. Furthermore, it is crucial to have control over the relative prominence of these two types of motion. Depending on the specific configuration, the resulting flow pattern can exhibit an *object-dominated* characteristic when an abundance of objects is incorporated, or conversely, a *self-motion-dominated* nature if fewer or no objects are present. By carefully manipulating these factors, ROFL enables the generation of optical flow patterns that faithfully capture the dynamics of real-world scenarios involving an interplay between self-motion and object motion.

## 13.1 Setup

Self-motion is meaningful if the observer is moving relative to a static background (i.e., the environment). We consider a solid background wall at a distance $Z = 1$ from the observer which extends to infinity in the $X - Y$ plane. Define the following coordinate frames

1. **Fixed coordinates**, represented with capital letters: $(X, Y, Z)$.
2. **Observer's coordinates**, represented with lowercase letters: $(x, y, z)$.

Both coordinate systems are centered around the observer. The difference is that $\hat{Z}$ is always perpendicular to the background plane, while $\hat{z}$ is defined such that it is parallel to the fixation point.

All points in the space are initially expressed in the fixed coordinates. For example, any given point like $\vec{P} = (X, Y, Z)$ has its coordinates defined in the fixed system. The same is true for fixation point, $\vec{F} = (X_0, Y_0, Z)$, and the observer's position $\vec{O} = (0, 0, 0)$, and so on. See Fig. 18.

### 13.1.1 Relating the two coordinate systems

How do we go from the fixed coordinates to the observer's coordinates? We need to find a rotation matrix that maps the $\hat{Z}$ direction to the direction defined by the fixation point. By definition, this will determine $\hat{z}$ and all the other directions in the observer's coordinates. First, let us express $\vec{F}$ using its polar coordinates:

$$\vec{F} = (X_0, Y_0, Z) \equiv \left( \|\vec{F}\|, \Theta_0, \Phi_0 \right), \tag{19}$$

where $\|\vec{F}\| = \sqrt{X_0^2 + Y_0^2 + Z^2}$, $\cos \Theta_0 = Z/\|\vec{F}\|$, and $\tan \Phi_0 = Y_0/X_0$. It turns out what we are looking for is a rotation by an amount $\Theta_0$ about a unit vector $\hat{u}$ that depends only on $\Phi_0$. That is,

$$\hat{u}(\Phi_0) = \begin{pmatrix} -\sin \Phi_0 \\ \cos \Phi_0 \\ 0 \end{pmatrix}, \tag{20}$$

and the rotation matrix is given by $R_{\hat{u}}(\Theta_0) = I + (\sin \Theta_0) L(\hat{u}) + (1 - \cos \Theta_0) L(\hat{u})^2$.

Here, $L(\vec{\omega})$ is the skew-symmetric matrix representation of a vector $\vec{\omega}$, that is

$$L(\vec{\omega}) := \vec{\omega} \cdot \vec{L} = \omega_X L_X + \omega_Y L_Y + \omega_Z L_Z, \tag{21}$$

where $L_i$ are the familiar basis matrices for $\mathfrak{so}(3)$, the Lie algebra of $SO(3)$:

$$L_X = \begin{pmatrix} 0 & 0 & 0 \\ 0 & 0 & -1 \\ 0 & 1 & 0 \end{pmatrix}, \quad L_Y = \begin{pmatrix} 0 & 0 & 1 \\ 0 & 0 & 0 \\ -1 & 0 & 0 \end{pmatrix}, \quad L_Z = \begin{pmatrix} 0 & -1 & 0 \\ 1 & 0 & 0 \\ 0 & 0 & 0 \end{pmatrix}. \tag{22}$$

The rotation matrix in its full form is expressed as

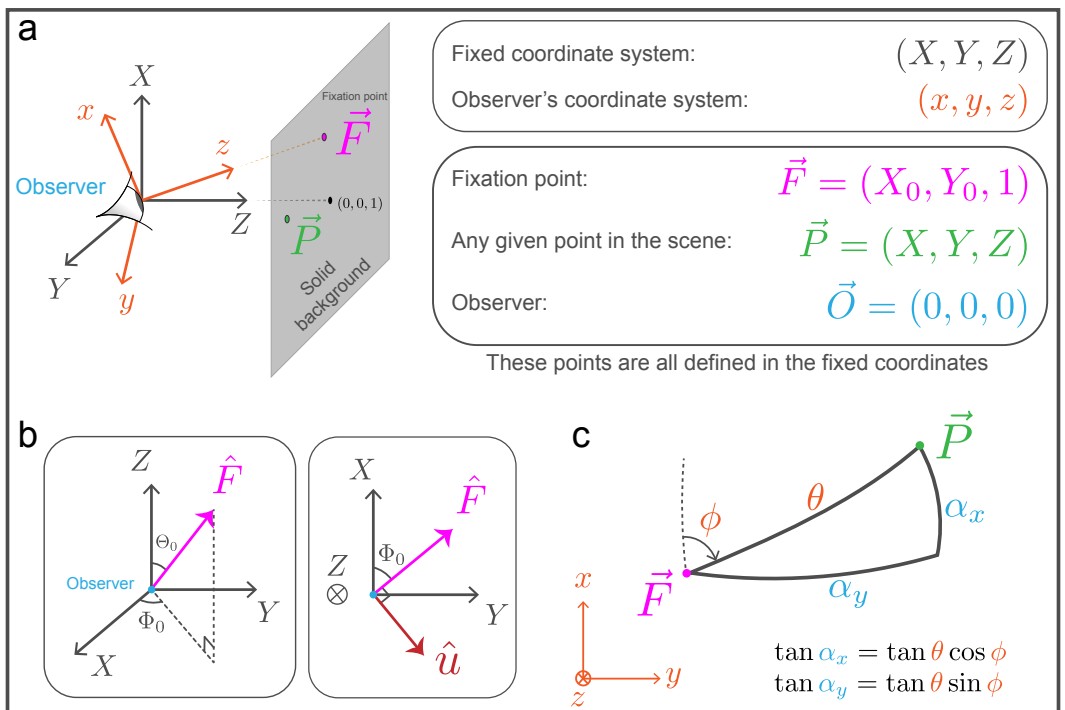

Figure 18: Setup. **(a)** Background and coordinate systems. **(b)** The rotation of the fixed coordinates $\left(\hat{X}, \hat{Y}, \hat{Z}\right)$ onto the observer-centric coordinates $(\hat{x}, \hat{y}, \hat{z})$ can be accomplished by applying a rotation of angle $\Theta_0$ around the unit vector $\hat{u}$ as defined in Equation 20. Note that by definition, $\hat{z}$ is always parallel to the fixation point. See the text for more details. **(c)** Perspective from the observer's point of view. The retinotopic coordinates of point $\vec{P}$ are denoted by $\alpha_x$ and $\alpha_y$, measured in radians.

$$R := R_{\hat{u}}(\Theta_0) = \begin{pmatrix} s^2 + c^2 \cos \Theta_0 & -cs\left[1 - \cos \Theta_0\right] & c \sin \Theta_0 \\ -cs\left[1 - \cos \Theta_0\right] & c^2 + s^2 \cos \Theta_0 & s \sin \Theta_0 \\ -c \sin \Theta_0 & -s \sin \Theta_0 & \cos \Theta_0 \end{pmatrix}, \tag{23}$$

where we have defined $c := \cos \Phi_0$ and $s := \sin \Phi_0$.

Any point $\vec{P} = (X, Y, Z)$ defined in the fixed coordinate system can be mapped to a point $\vec{p} = R^T \vec{P}$ with components $\vec{p} = (x, y, z)$ given in the observer's system. For example, it can be shown that $R^T \vec{F}$ is parallel to $\hat{z}$, which is how we defined $R$ in the first place. See Fig. 18.

In conclusion, we have a rotation matrix $R$ that is fully determined as a function of the fixation point $\vec{F}$. Applying $R$ rotates the fixed coordinate system onto the observer's system

$$\begin{aligned} \hat{x} &:= R\hat{X}, \\ \hat{y} &:= R\hat{Y}, \\ \hat{z} &:= R\hat{Z}. \end{aligned} \tag{24}$$

This way, the observer's coordinates $(x, y, z)$ are obtained ensuring that the fixation point is always parallel to $\hat{z}$.

### 13.1.2 Adding motion

Consider a point $\vec{P} = (X, Y, Z)$. In the previous section, we showed that the rotation matrix in Equation 23 can be used to relate $\vec{P}$ to its observer-centric coordinates

$$\vec{P} = R\vec{p}. \tag{25}$$

Differentiate both sides to get

$$\vec{V} := \frac{d\vec{P}}{dt} = \frac{dR}{dt}\vec{p} + R\frac{d\vec{p}}{dt}, \tag{26}$$

where $\vec{V}$ is the velocity of the point relative to the observer, with components measured in the fixed coordinate system.

### 13.1.3  Derivative of the rotation matrix

Here we will derive a closed-form expression for $dR/dt$ in terms of the observer's angular velocity. This is important because it essentially captures the effects of eye movement.

Let us use $\vec{\omega} = (\omega_X, \omega_Y, \omega_Z)$ to denote the angular velocity of observer's coordinates, with components expressed in fixed coordinates. The time derivative of $R$ is given as follows

$$\frac{dR}{dt} = L(\vec{\omega})R, \tag{27}$$

where $L(\vec{\omega})$ is given in Equation 21. Taken together, equations 3, 4, 8, and 9 yield

$$\begin{aligned}
\frac{d\vec{p}}{dt} &= R^T\vec{V} - \left[R^T L(\vec{\omega})R\right]\vec{p} \\
&= R^T\vec{V} - \left[R^T \begin{pmatrix} 0 & -\omega_Z & \omega_Y \\ \omega_Z & 0 & -\omega_X \\ \omega_Y & \omega_X & 0 \end{pmatrix} R\right]\vec{p}
\end{aligned} \tag{28}$$

This equation allows relating the velocity of a moving point expressed in fixed coordinates, $\vec{V}$, to its observer-centric equivalent, $\vec{v} := d\vec{p}/dt$. Importantly, Equation 28 is a general relationship valid for any combination of velocities and rotations.

### 13.1.4  Motion perceived by the observer

Before discussing specific applications, we need to solve one remaining piece of the puzzle: what is $\vec{v}$, and how is it related to the motion perceived by the observer? To address this, we need to map the retinotopic coordinates of the observer (measured in radians), to points in the environment (measured in the observer-centric coordinates).

Denote the retinotopic coordinates using $(\alpha_x, \alpha_y)$. The goal is to relate each $(\alpha_x, \alpha_y)$ pair to a point in the world $\vec{p}$ with coordinates $(x, y, z)$. It can be shown that

$$\begin{aligned}
\tan\alpha_x &= x/z = \tan\theta\cos\phi, \\
\tan\alpha_y &= y/z = \tan\theta\sin\phi,
\end{aligned} \tag{29}$$

where $\theta$ and $\phi$ are the polar coordinates of $\vec{p}$. See Fig. 18c for an illustration.

Differentiate both sides of Equation 29 and rearrange terms to obtain the time derivative of $(\alpha_x, \alpha_y)$ as a function of $\vec{v} = (\dot{x}, \dot{y}, \dot{z})$ and $\vec{p}$. That is,

$$\begin{aligned}
\dot{\alpha}_x &= \frac{\dot{x}z - \dot{z}x}{z^2 + x^2} \\
\dot{\alpha}_y &= \frac{\dot{y}z - \dot{z}y}{z^2 + y^2}
\end{aligned} \tag{30}$$

Thus, we obtained a relationship between retinotopic velocity $(\dot{\alpha}_x, \dot{\alpha}_y)$ (measured in $radians/s$) and velocity of objects in the world (measured in $m/s$). In conclusion, equations 28 and 30 enable us to compute the observer's perceived retinotopic velocity for any type of motion.

## 13.2 Example: maintaining fixation

We are now ready to apply the formalism developed in the preceding section to create optical flow patterns in realistic settings. In this section, we will consider a specific example: self-motion while maintaining fixation on a background point. This is an important and ecologically relevant type of motion, as we often tend to maintain fixation while navigating the environment—think about this next time you are moving around. In fact, one can argue that fixation occurs more naturally during behaviors such as locomotion, as opposed to typical vision experiments where the animals are stationary and passively viewing a screen.

What does it mean to maintain fixation while moving? It means the eyes rotate to cancel out the velocity of the fixation point. Assume the observer is moving with velocity $\vec{V}_{self}$. Equivalently, we can imagine the observer is stationary while the world is moving with velocity $\vec{V} = -\vec{V}_{self}$ (this is the same $\vec{V}$ that appears in Equation 28).

Any eye movement can be represented mathematically as a rotation of the observer's coordinates with respect to the fixed coordinates. Use $\vec{\omega}$ to denote the angular velocity of the rotation. We can determine $\vec{\omega}$ by requiring that the rotation cancels out the velocity of the fixation point. To this aim, we first compute the normal component of $\vec{V}$ with respect to the fixation point $\vec{F}$ as follows

$$\vec{V}_{\perp} := \vec{V} - \frac{\vec{V} \cdot \vec{F}}{\|\vec{F}\|^2} \vec{F}. \tag{31}$$

Use the above equation to compute the angular velocity

$$\vec{\omega} = \frac{\vec{F} \times \vec{V}_{\perp}}{\|\vec{F}\|^2}. \tag{32}$$

Plug Equation 32 in Equation 28 to find $\vec{v}$, and use Equation 30 to find $(\dot{\alpha}_x, \dot{\alpha}_y)$. This way, we just calculated the motion perceived by the observer during self-motion while maintaining fixation.

## 13.3 The algorithm

Start by choosing a fixation point $\vec{F} = (X_0, Y_0, Z)$, field-of-view extent ($fov$), and retinal resolution ($res$). For example, $fov = 45°$ and $res = 5.625°$. Use this to create a mesh grid for $(\alpha_x, \alpha_y)$. The grid covers $(-fov, +fov)$ degrees in each dimension and has shape $2 * fov/res + 1$, or $17 \times 17$ in our example. Use Equation 29 to compute $(\theta, \phi)$ pair for each point on the $(\alpha_x, \alpha_y)$ grid.

Because the simulation is scale-invariant, we can always fix the distance between the observer and the background wall. Assume $Z = cte. = 1$, and use equations 23 and 25 to find $\vec{p} = (x, y, z)$ per grid point. Note that we assume infinite resolution for the points in the environment.

So far, we partitioned the retina into a grid and found the observer-centric coordinates of a point in the real world that falls on each grid point. Now we can sample a random self-motion velocity $\vec{V}_{self}$ and use the results described in the previous sections to find $(\dot{\alpha}_x, \dot{\alpha}_y)$. This provides us with the $2D$ retinal velocity vector on the grid and concludes the algorithm for the simple case of `fixate-0` (i.e., no objects).

Please refer to our code for additional details, including instructions on how to incorporate independently moving objects into the scene.

### 13.3.1 Valid fixation points

The observer-wall distance is constrained to be a positive value. That is, the observer is always situated on the left side of the wall ($Z > 0$ for all points; Fig. 18). This constraint, combined with a given $fov$ value and Equation 23 results in a theoretical bound on which fixation points are valid. To derive this bound, we start by considering the $\hat{Z}$ component of Equation 25

$$(R\vec{p})_3 = R_{31}x + R_{32}y + R_{33}z = Z > 0. \tag{33}$$

Divide both sides by $z$ and use Equation 29 to get

$$R_{31} \tan \alpha + R_{32} \tan \beta + R_{33} > 0. \tag{34}$$

Plug the matrix entries from Equation 23 and rearrange to get

$$\begin{aligned} \sin \Theta_0 (\cos \Phi_0 \tan \alpha + \sin \Phi_0 \tan \beta) &< \cos \Theta_0, \\ \Rightarrow \quad X_0 \tan \alpha + Y_0 \tan \beta &< Z. \end{aligned} \tag{35}$$

For a given $fov$ value, we have $-\tan(fov) \leq \tan \alpha, \tan \beta \leq \tan(fov)$. Assuming $Z = 1$, we find that the $(X_0, Y_0)$ coordinates of a valid fixation point should satisfy the following inequality

$$|X_0| + |Y_0| < \frac{1}{\tan(fov)}. \tag{36}$$

In other words, the $L_1$ norm of $(X_0, Y_0)$ should be less than $1/\tan(fov)$. For example, when $fov = 45°$, the upper bound is 1. This result can be used to choose an appropriate $fov$: smaller values allow a larger phase space to explore for fixation points.

## 13.4 Conclusions

We introduced Retinal Optic Flow Learning (ROFL), a novel synthetic data framework to simulate realistic optical flow patterns. The main result is given in Equation 28, which shows the simulation has the potential to generalize to various other types of motion. For instance, the observer might engage in smooth pursuit of a moving object instead of maintaining fixation, a scenario that can be modeled simply by adjusting the choice of $\vec{\omega}$ in Equation 28. This flexibility enables us to generate a diverse repertoire of realistic optical flow patterns, which can be used in training and evaluating unsupervised models, among other applications.

We believe that our framework will prove valuable not only to neuroscientists studying motion perception but also to machine learning researchers interested in disentangled representation learning. By bridging these domains, we aim to foster interdisciplinary collaboration and advance our understanding of both biological and artificial information processing systems.

