# OpenReview forum: "Hierarchical VAEs provide a normative account of motion processing in the primate brain"
_NeurIPS.cc/2023/Conference — NeurIPS 2023 poster_

### Official Review · Reviewer_nG5y · 2023-07-02

**Soundness:** 2 fair
**Presentation:** 3 good
**Contribution:** 2 fair
**Rating:** 4
**Confidence:** 5

**Summary:**

- This empirical study develops a new framework for neural representation of motion in cortex.
- The authors propose a novel stimulus synthesis method for model training, parametrically generating optic flow fields w/ low-dim latent structure, rather than using pixel-space (i.e. image-computable) inputs.
- They modify previously-developed hierarchical Variational Autoencoders, and argue that the inductive architectural biases of such networks produce a more desirable representation of motion, as quantified via entanglement metrics and neurophysiological alignment via linear regression.
-  Apart from the stimulus synthesis method, which I believe to be well-motivated and a sensible approach to modeling the inputs, I believe the theoretical advance of this study is marginal, and am unconvinced that the proposed architecture that is central to the paper provides a significant advancement to our understanding of cortical motion processing.

UPDATE: Sep 1, 2023. I have read the rebuttal, my issues with the paper remain, and I maintain my score.

**Strengths:**

- I enjoyed the proposed stimulus synthesis method for training, and the last figure teasing apart contributions of the different generative factors to performance is nice.
- Writing is mostly clear and easy to follow (clarity suggestions below).
- The use of hierarchical VAEs to producing more disentangled/unentangled motion representations is interesting.

**Weaknesses:**

- The logic surrounding the physiological alignment analyses feels unsettlingly circular and/or redundant. This perhaps applies to Higgins' paper too since they do similar analyses. 1) The cNVAE is shown to produce improved DCI scores over vanilla VAE. 2) Your assessment of alignment effectively penalizes models with worse disentanglement  (i.e. a more distributed code), and therefore adjudicates that the cNVAE has better alignment with physiological data.
- In fact, a central motivation of this study relies on a conjecture that disentanglement and/or unentanglement are normative goals of the dorsal stream. While this objective would be appealingly analogous to those proposed in the object recognition and ventral visual stream literature, I am unconvinced that this is the case given the evidence presented here.
- The above points are compounded by the fact that, as you mentioned in Fig. 7, the non-hierarchical VAE architecture does comparably on predicting MT responses to the cNVAE.

**Questions:**

- The paragraph introducing the architecture (L108) remains opaque to me. It would be helpful to include a more thorough description of the dimensionality of x, z1, z2, z3. Are you concatenating z1,2,3 to form your 420D cNVAE latent, and comparing it to a single 420D z VAE latent? This is what I am inferring from Fig. 3.
- There are many acronyms that are introduced and never explicitly written out. I am familiar with much of the VAE and disentanglement literature but the general reader shouldn't have to be. E.g. NVAE, DCI, TCVAE (since this one isn't pertinent to your study you can just remove it altogether instead of introducing more terms to the reader), .
- Please elaborate on what end-point error is (L137).
- Please include a sentence on what permutation importance is; I had to look up the sklearn docs.
- VAEs are inherently stochastic neural networks with exact mean and covariance structure available to the experimenter. However, covariance is typically thrown out most studies using VAEs, including this one. This is at odds with the fact that noise exists and neural coding is impacted by noise (e.g. the classic Averbeck 2006 review on neural correlations).  You mention there are repeats in some trials of the CRCNS dataset; is it possible to quantify whether or not your model captures the stochastic structure of the neural responses?

**Limitations:**

There was discussion of limitations.

---

> ### Author Rebuttal · Authors · 2023-08-10
>
> First, we want to contextualize all “weaknesses” about the brain alignment score (Fig. 8) in the greater scope of the manuscript. Fig. 8 is one of the 5 metrics we use to evaluate the learned representations (the others being untangling, disentanglement, completeness, and neural prediction). All 5 metrics favor the cNVAE. Therefore, this is a small fraction of the results.
>
> > The logic surrounding the physiological alignment...
>
> We interpret your comment to mean our alignment metric favors disentangled codes, therefore any disentangled code will better align with MT (or any other representation, for that matter).  Although we agree that our alignment score is far from perfect, there are several reasons why we believe this is not a weakness:
>
> 1. Maybe “redundant” but not “circular”: If the alignment metric was biased to favor disentangled codes, this would only make it redundant because we do not start from a motivation that codes should be disentangled (more below).
>
> 2. There is no such thing as a *universally* disentangled code, and therefore this alignment score would have to depend on what is being disentangled. In its most common definition, disentanglement depends explicitly on how we define ground truth variables. We explain this point in the appendix (section: “disentanglement is in the eyes of the beholder”; Fig. S8) and show that disentanglement scores change with different ground truth definitions.
> 3. The disentanglement score for the cNVAE is not that different from vanilla VAE (Fig. 5). Where the models differ most is in untangling and completeness. Given this, we think it is unlikely that disentanglement is driving the high MT alignment.
> 4. Our alignment score (Fig. 8) is more dependent on the hierarchy than DCI metrics, as indicated by the good performance of the cNAE.
> 5. We believe *untangling* is the most important feature of cNVAE, and where it shines (Fig. 4b). Despite their similar names, “untangling” and “disentanglement” are independent concepts, where “untangling” simply means the ground truth factors are linearly decodable ([DiCarlo & Cox, 2007](https://tinyurl.com/dicarlocox)). Although several recent lines of work have argued that disentangled codes are desirable, others have argued that distributed codes are good as long as they support untangling ([Rigotti et al., 2013](https://tinyurl.com/rigotti13)).
> 6. The reviewer is holding us to a standard that no prior work is held to. When it comes to prediction performance, the best model is cNVAE (at $\beta=0.8$) with an improvement of 0.008 over the best VAE (see Table 3). In comparison, Mineault et al. (ref 29, which was a spotlight NeurIPS paper) report a minuscule improvement of 0.001 to select their best model.
>
> We completely agree with the reviewer that our measure of alignment is not perfect and we highlighted limitations of prediction and alignment in our main text and supplemental material. We think the reviewer is correct to be dissatisfied with disentanglement and linear regression as a measure of brain alignment. There is a lot of future work to do before the field converges on metrics for evaluating the similarity of pairs of representations, as highlighted in recent work (e.g., [Han et al. 2023](https://tinyurl.com/hanicml23)). A potential route forward is geometric analyses ([Williams et al., 2021](https://tinyurl.com/awshape21)). Ultimately, we don’t think disentanglement is central to our results, but is one of several metrics that can be used to evaluate a code. Because Fig. 8 is a small part of our results, we are happy to move it to the supplemental material and modify the text accordingly if the reviewer feels we overstated its importance.
>
> > In fact, a central motivation of this study...
>
> That was not a motivation for this study. Our study was motivated by conjectures from Helmholtz and Mumford (see our global rebuttal). In our efforts to compress our introduction to fit the page requirements, we collapsed this idea too much and this probably misled the reviewer. We will fix this in our revisions and address this point directly here.
>
> We were primarily motivated by the idea that hierarchical inference is important for representation learning, which we evaluated using a number of metrics that have been proposed by neuroscientists and ML researchers and concluded that hierarchical inference leads to several improvements. Our approach contrasts with previous work on dorsal stream (e.g., Mineault et al. 2021) where linearly decoding ground truth was literally the objective (they used supervised ResNets). Our models were trained solely using the standard ELBO or beta-VAE loss. Our results demonstrate that unsupervised models w/ hierarchical inference untangle more than models w/o hierarchy. Importantly, untangling and disentanglement are metrics we use to evaluate the role of hierarchy in learned representations, and they were neither a motivation nor objective.
>
> > The above points are compounded…
>
> Our best model was a cNVAE by 0.008 over the best VAE, which is 8 times larger than the selection criterion used by the SOTA model of the dorsal stream (Mineault et al., 2021), which we outperform by over a factor of 2.
>
> > VAEs are inherently stochastic…
>
> We agree there is an interesting parallel between the covariance in VAEs and the covariance of neurons in the brain. One practical reason why we cannot examine this is our dataset consists of single-unit recordings, making it impossible to compare the covariance of these neurons to VAEs.
>
> Interestingly, we did look at the covariance of latent representations and found that cNVAE encodes its input in a much larger dimensionality than the vanilla VAE (Figure R1-f in the global rebuttal). What information is encoded in those correlations? Is it related to noise correlations? We don’t know, but we like this suggestion and plan to study this later with a population-recording dataset.
>
> > Questions
>
> We will implement the remaining suggestions in our revisions.

---

> > ### Comment · Reviewer_nG5y · 2023-08-11
> >
> > Thanks for your responses to my comments, and the preliminary changes you wrote in your global rebuttal.
> >
> > I was not aware that Mineault et al. [29] received a spotlight for their paper; however, I don't think this is constructive to the discussion (I notice you also mention it in another reviewer's rebuttal), nor would it be appropriate for me as a reviewer to factor that into my judgment of the present study.
> >
> > I can appreciate that your average prediction score is higher than [29]. I was mainly bringing attention to your statement in both the main text and fig 7 that cNVAE and VAE performance are comparable (indeed, the standard errors overlap in the table). This leads me to question how effective cNVAE and its inductive biases are to motion processing, when prediction performances of these models overlap.
> >
> > I'm inclined to raise my score from 3 -> 4 for the clarity changes you promised. But, provided the evidence in your study, I remain unconvinced that untangling, as emphasized in the paper and your rebuttal, are imperative to neural coding of motion.

---

> > > ### Author Response · Authors · 2023-08-11
> > >
> > > Thank you for taking the time to read our responses and for engaging with us.
> > >
> > > We genuinely appreciate your effort in delving into our work and sharing your thoughts. Your insightful critique of our work prompted us to think carefully about our contributions. As a result, we included some of those clarification points in our global rebuttal and will add more to our revised manuscript. We were pleased to learn that you find our efforts to enhance clarity in line with your expectations.
> > >
> > > We believe there might be some remaining misinterpretations that we would like to address in a more transparent manner below.
> > >
> > > > I remain unconvinced that untangling, as emphasized in the paper and your rebuttal, are imperative to neural coding of motion
> > >
> > > We did not set out to test whether untangling is imperative to cortical motion processing, nor do we claim that. We were inspired by the hypothesis-driven approach of testing for the presence of particular information with a decoder (e.g., see [Kriegeskorte & Diedrichsen, 2019](https://www.annualreviews.org/doi/abs/10.1146/annurev-neuro-080317-061906)): if an [artificial or biological] information processing system represents a feature of its sensed inputs, then those features should be easily decodable from the representation.
> > >
> > > As stated in our rebuttal, our objective was to investigate the role of hierarchical inference in learned representations. We were inspired by longstanding conjectures in neuroscience that representations should explicitly represent generative causes of the senses.
> > >
> > > Motivated by this hypothesis, we set out to test whether VAEs w/ hierarchy do better in untangling compared to those w/o hierarchy. Our results provided strong evidence in favor of that.
> > >
> > > In a separate set of experiments, with a totally different motivation, we asked whether the hierarchical models are also more “aligned” with biological representations. To test this, we used the existing MT dataset from Mineault et al. [29] and found that not only were we able to outperform SOTA with over 2x gain, but also our cNVAE was better at predicting MT neurons, although marginally.
> > >
> > > > I can appreciate that your average prediction score is higher than [29]
> > >
> > > We are grateful for your acknowledgment of the substantial performance gain observed in our study compared to [29]. We believe this achievement stands as a significant contribution in its own right, offering valuable insights into the potential range of attainable performances on the MT data.
> > >
> > > In sum, we hope that these added insights will contribute to a clearer understanding of our motivations and conclusions. If you come across any specific instances in our manuscript that might not align with this clarification, please let us know. We are more than happy to make any necessary language adjustments to ensure that our motivations and conclusions are accurately reflected.

---

> > > > ### Comment · Reviewer_nG5y · 2023-08-14
> > > >
> > > > Thanks and I appreciate your added response.
> > > >
> > > > There still remains the fact that the central theme is on the importance of hierarchical inference in learned motion representation in the primate brain, while vanilla VAEs' predictive performance are statistically indistinguishable (re: error bars in the table 3 and fig 7) from the proposed architecture.
> > > >
> > > > With this in mind, as well as having gone through the paper again, and the listed pertinent references during the review phase: I believe it's possible there is a revised form of this manuscript that presents a clearer, more convincing version of your story. But, this undoubtedly would be different enough from its current state (in line with Reviewer Y46x's most recent statement on there already being many promised changes), to have a chance of affecting the overall narrative. Thus, I am not inclined to further increase my score.

---

> > > > > ### Author Response · Authors · 2023-08-18
> > > > >
> > > > > Thank you for the comment.
> > > > >
> > > > > > There still remains the fact that the central theme is on the importance of hierarchical inference in learned motion representation in the primate brain
> > > > >
> > > > > As discussed in previous responses, the electrophysiology results constitute a fraction of the measures we apply to assess the role of hierarchy in our model and follow current work in the field.
> > > > >
> > > > > > while vanilla VAEs' predictive performance are statistically indistinguishable (re: error bars in the table 3 and fig 7) from the proposed architecture.
> > > > >
> > > > > Although the predictive power of cNVAE and VAE overlap, several recent studies have suggested predictive power alone is not sufficient to conclude anything strongly about alignment (e.g., see [Han et al., 2023](https://proceedings.mlr.press/v202/han23d.html) for a demonstration of this point), which is in part why we compare models using multiple metrics. More broadly, the factors resulting in alignment between pairs of representations is an important open problem in the field (please refer to our rebuttal above for details and citations). As a result, we believe the extreme sparsity of latent-to-neuron relationships is interesting in itself and warrants a thorough exploration in future work while using data from more brain areas, as also discussed by us.
> > > > >
> > > > > Overall, our results suggest that the latent-to-neuron sparsity is more strongly an architectural effect compared to other factors such as loss function. To appreciate this point, please compare cNAE with AE in Figure 8. This result together with the preceding figures leads us to conclude that the hierarchical architecture of cNVAE is the primary reason driving its superior performance, as revealed by the application of various metrics and evaluations included in the present work.
> > > > >
> > > > > > I believe it's possible there is a revised form of this manuscript that presents a clearer, more convincing version of your story. But, this undoubtedly would be different enough from its current state (in line with Reviewer Y46x's most recent statement on there already being many promised changes), to have a chance of affecting the overall narrative. Thus, I am not inclined to further increase my score.
> > > > >
> > > > > We are grateful for the comments from this reviewer and all others, which gave us a great roadmap of what aspects did not come across clearly, and we hope our explanations in the rebuttal went some way in detailing these changes. Because this has *clarified* rather than *changed* the results, we do not view the outcome as a different paper. We believe the revised manuscript can still be judged on the merits of the existing results included in our submission. Due to the shortness of the rebuttal, we have not quoted all the revisions in full detail, but encourage the reviewer to ask for anything specific that might be score-driving so that they might better judge the changes.

---

### Official Review · Reviewer_ZQpT · 2023-07-04

**Soundness:** 2 fair
**Presentation:** 2 fair
**Contribution:** 3 good
**Rating:** 7
**Confidence:** 4

**Summary:**

The paper introduces a synthetic data framework called Retinal Optic Flow Learning (ROFL) and uses that framework to test the performance of unsupervised models on two learning tasks: reconstructing grand truth variables and predicting the response of MT neurons. By imposing a latent hierarchical structure, the authors observed improvements along three axes: on the linear decodability of the ground truth, on the predictability of MT neuron responses and, finally, on identifying the causal structure of the world as a major factor driving these results.

**Strengths:**

As far as I understood (given the lack of clarity on a few aspects regarding the problem definition and related work, please look at my comment below), the authors  introduced a method to generate synthetic data and then modeled the generated data using an ensemble of model architectures of their choice. While I am dubious of the novelty and the sensibility of the idea, the experimental results presented seem to validate the argument. Another important strength of the paper is that the authors are pretty open and clear about the weaknesses of their method, which is something to comment them for.

**Weaknesses:**

I am dubious of the idea because it seems that the authors generate synthetic data and then choose an ensemble of architectures to model them. Also, it is hard to understand what is the main problem tackled by the paper, how this work relates to the state of the art and most importantly what is the improvement and contribution compared to the SOA.

There’s no Related Work section in the manuscript and the method is not compared against any other methods in the experiments. Are there many papers that deal with the same problem? Reference 29 seems to be one such paper and it is compared against in the experiments. However, given that this reference has a different dataset and model, I see no common ground for comparison. And given the peculiar structure of this problem area (generating data and then modeling them), the authors need to find way to highlight their contributions.

All these make it hard to assess the paper. I recommend the authors add a related work section, clearly connecting their paper with the state of the art, and clarifying the novelty and contribution.

In you related work section, you may want to consider referencing some papers that use hierarchical latent spaces for real-world data, such the following one (which is a newly published paper, so it’s understandable that you hadn’t included it).

Generative Decoding of Visual Stimuli, https://openreview.net/pdf?id=57OuafQmu8

**Questions:**

“Importantly, this framework allows us to manipulate both the architectures of the models and the causal structure of the world they are trained on” -> As far as I understood, you chose the models and the method for generating synthetic data. Being able to manipulate the architecture and structure of data is somehow expected. Why is that then an important fact and statement to make? If it’s not, please consider removing it.

“We found that a single inductive bias, hierarchical latent structure, yields several improvements.” -> What is the metric that you use to assess performance and what do you compare against? The metrics are only qualitatively described in the paper. Please consider condensing that information in a clearly defined section, something like “Metrics and Baselines”, and give a clear, quantitative definition of the metrics.

“the brain engages in hierarchical Bayesian inference ” -> Based on what I read, I do not believe you have presented sufficient evidence to back up such a strong claim. The experimental study on predicting MT response does not suffice. If you meant to back the claim using more of the results you presented please make it more clear in the main manuscript or else remove it.

The metrics that are used in the experimental section such as informativeness, disentanglement and completeness are not clearly defined. There are some papers cited where those metrics are mentioned in the text and I am sure that someone would find the definition if they look them up. However, given that these metrics are not widely used, I think the authors should give the definition in the text.

What are the details of the data that you used for the MT learning task? There’s little to no information regarding what these data are, how they are collected, pre-processed etc. I understand that you cite a couple of papers that probably have these details, however, I think you have to give some details and explain how you use this dataset.

What is the learning task for the MT prediction, i.e., what is the quantity you’re predicting? Is it the firing rate shown on figure 6? If so, how is that related to the hierarchical modeling in the previous section? Details on the dataset would help clarify.

What is the AirSim dataset that the authors compare against in the MT prediction task? And how does it related to the newly introduced ROLF? If it’s a completely different synthetic data framework I do not see the point of comparing against it.

Is the label in Figure 3 correct? There’s a reference to Figure 2. Maybe the author meant to refer to the bottom of Figure 3.

On lines 225-227 you say “all metrics for a broad range of β 227 values (Figure 5).” I am assuming that by all metrics you mean the  “untangling”, “disentengling”, and “brain-alignment” mentioned in line 195. However, the labels on figure 5 are slightly different, “Informativeness”, “Disentanglement” and “Completeness”. Please clarify what the metrics are and quantitatively define them, as they are not well known metrics.

Why the word “learning” in ROFL? There is no learning involved in the data generation, as far as I understood.

**Limitations:**

The authors have clearly and openly discussed the limitations of their work.

---

> ### Author Rebuttal · Authors · 2023-08-10
>
> > As far as I understood…comment them for.
>
> We struggled with how to answer this review. It seems that comments alternate between not understanding or knowing the relevant literature, and asserting (confidently) that the work was not novel and/or incorrect. Furthermore, in several cases, we already have whole sections devoted to some of the reviewer's concerns.
>
> As a result, we hope that we can largely address these concerns by clarifying our contributions, and have reworked parts of the text accordingly. Along these lines, we have included a draft Related Work section in the Global Rebuttal, and this should clarify where this paper sits in the existing literature. We have also reiterated the main points in our global response to reviewers and addressed individual ones here. We hope this clarifies the “sensibility” of the ideas, which stem from two major conjectures in neuroscience.
>
> > I am dubious of the idea…
>
> We were motivated by the idea that hierarchical inference is important for learned representations. This is an old idea in neuroscience and a newer idea in ML. To evaluate the role of hierarchical inference in learned representations, it was essential to generate synthetic data with ground truth variables. We then evaluated the representations using a number of metrics and concluded that hierarchical inference leads to several improvements using metrics that have been proposed by neuroscientists and ML researchers.
>
> In addition to our main goals, our paper moves beyond solely evaluating the reconstruction performance of hierarchical VAEs. As far as we know, a comprehensive investigation of representation learning in hierarchical VAEs has not yet been done: this is the first.
>
> > …and the method is not compared against any other methods in the experiments. Are there many papers that deal with the same problem?
>
> We compare directly to a recently established benchmark, which we cite extensively: ref [29] – which was a spotlight NeurIPS paper in 2021. We are the only other paper to use this benchmark thus far.
>
> > Reference 29 seems to be one such paper…no common ground for comparison.
>
> We use the same MT dataset as Ref 29 (crcns-mt1), and the model is indeed different which we now detail in the Related Work.
>
> > And given the peculiar structure of this problem area …
>
> Synthetic data with ground truth factors is essential to interrogate the learned representations and is standard practice in disentangled representation learning: https://paperswithcode.com/task/disentanglement
>
> Our manuscript had an entire section on this point (specifically, section 2.1 “Using synthetic data to test hypotheses about the causal structure of the world”)
>
> > In your related work section, you may want to consider referencing...
>
> Thank you for this ref. We now address this (and other) papers in the Related Work section. We note here that Miliotou et al. have a very different goal than ours: to decode images from fMRI voxel activations (using a hierarchical VAE); whereas, our work focuses on learned representations and how they depend on architecture (hierarchical vs. non-hierarchical), loss function (variational vs. maximum likelihood), and the generative factors of variation in the training set.
>
> > Q1: As far as I understood...
>
> Briefly, our ability to manipulate the structure of the data (and model) is exactly why it is useful for our work. We highlight this point at multiple places in the manuscript, including in the Introduction and discussion (line 290). We even include a section about it (section 2.1 “Using synthetic data to test hypotheses about the causal structure of the world”).
>
> > Q2: What is the metric that you use to assess performance…
>
> Because our focus was on the learned representations, the definition of the performance metrics based on the reconstruction loss was relegated to the Appendix (Fig. S5 shows a summary). We already did state in the Introduction (line 54) what we meant by “improvements”. We will add a table summarizing reconstruction performance and additional text describing the metrics computed.
>
> > Q3: “the brain engages in hierarchical Bayesian inference "...
>
> Here we agree. Our observations are consistent with this notion, rather than prove it. We will modify the language accordingly.
>
> > Q4: The metrics that are used in the experimental section...
>
> These metrics are widely used in the VAE literature, but perhaps not more generally. We will add more details about these metrics in the revised text.
>
> > Q5: What are the details of the data...
>
> We will add some key details to the main paper, and refer the reader to the supplemental where we include every detail to make this work completely explained given the length constraints of the main text.
>
> > Q6: What is the learning task for the MT prediction...
>
> We follow standard neural modeling approaches and predict the binned spike count of each neuron, which when normalized by the time bin size, results in a “firing rate” (number of spikes per second).
>
> > Q7: What is the AirSim dataset…
>
> Both ref 29 and our manuscript analyze MT data from the same CRCNS dataset. We both train models using completely separate synthetic datasets that we then use to predict MT. Ref 29 used AirSim to train supervised 3D ResNets that they then use to predict MT (using a completely different stimulus). We created ROFL to simplify the causal structure of motion in the world and then, as a part of our analysis of the learned representations, we predict MT. Importantly, we get a factor of 2x performance gain over the previous best model (ref 29).
>
> > Q8: Is the label in Figure 3 correct?
>
> Yes.
>
> > Q9: On lines 225-227 you say...
>
> “Untangling” and “informativeness” are the same which we discuss in lines 197-202 and also in supplemental section 1.4.
>
> > Q10: Why the word “learning” in ROFL?
>
> The dataset is for unsupervised learning of latents: thus we consider this to be a “learning framework” overall.

---

> > ### Comment · Reviewer_ZQpT · 2023-08-16
> >
> > Thank you for the rebutal and appologies for not responsing earlier.
> >
> > Even though I still remain dubious of the idea, I slightly increased my score to 6. With the changes that the authors said they'll do, I think that the paper would be in a much better shape. The most important one that I'd like to see is a clearly defined Related Work section.
> >
> > Also, please add details and clearly define all the metrics. Regarding Figure 5, it is still not clear what "Completeness" is.

---

> > > ### Author Response · Authors · 2023-08-17
> > >
> > > Thank you for your comments and questions. Below we address the remaining items.
> > >
> > > > Also, please add details and clearly define all the metrics.
> > >
> > > The camera-ready version will contain a Table and an associated text section that defines and explains the “Metrics and Baselines” used in this work, as originally suggested by the reviewer, which we thank them for.
> > >
> > > > Regarding Figure 5, it is still not clear what "Completeness" is.
> > >
> > > Completeness measures the average number of latent variables $z_i$ required to capture any single ground truth variable $g_j$. If a single latent contributes to $g_j$’s prediction, the score will be 1 (complete). If all latent variables equally contribute to $g_j$’s prediction, the score will be 0 (maximally overcomplete).
> > >
> > > It is noteworthy that the completeness score has also been called *compactness* ([Ridgeway & Mozer, 2018](https://proceedings.neurips.cc/paper_files/paper/2018/hash/2b24d495052a8ce66358eb576b8912c8-Abstract.html)). For more info, please see the original DCI paper ([Eastwood & Williams, 2018](https://openreview.net/forum?id=By-7dz-AZ)) or a recent extension of it ([Eastwood et al., 2023](https://openreview.net/forum?id=462z-gLgSht)).
> > >
> > > In the final version of our manuscript, we will clarify what each metric measures in the main text, and add this background information about metrics utilized in our study including mathematical formulas used to compute the scores, in the “Metrics and Baselines” section in the supplemental.
> > >
> > > We hope that this will properly address the reviewer's concerns and comments.

---

### Official Review · Reviewer_Y46x · 2023-07-06

**Soundness:** 3 good
**Presentation:** 2 fair
**Contribution:** 3 good
**Rating:** 5
**Confidence:** 3

**Summary:**

The authors present a framework to evaluate motion detection in different DNNs. First, they intoduce a new concept to create flowfields for optical sitmuli, which include local and global motion and additionally fixation points. They use the parametrized stimuli to train a new hierarchical VAE (cNVAE) and compare it to other DNNs in terms of different disentanglement metrics and similarity to neural recordings of MT. cNVAEs outperform standared VAEs in the disentanglement setting, and are comparable to VAEs for neuronal prediction while their latent is more aligned with single neural recordings. Training on different synthetic flowfield datasets showcase the influence of the treaining data to predict neural respones.

**Strengths:**

Clear description of stimulus genearation, and overall a solid paper.

Clear and detailed description of the limitations.

Interesting hierarchical VAE model which could be transferred to other learning tasks and brain areas.

**Weaknesses:**

**Major**
- Evaluation: While the authors do a good job in a high-level evaluation of the models (applying different evaluation scores and comparison to other models), further in-depth investigation would strengthen the paper significantly.
For example, it would be interesting to investigate: the learned representation (manifold); the necessary dimension of the latent space, and how the performance and disentanglement change with different dimensions; the hierarchical structure (in the model and the neural data); counterexamples where the model fails and why it fails; further investigation of the receptive fields for different neurons; some of them are touched on in the Appendix, but a clear link to the main text is missing.
- The paper would benefit from some restructuring to disentangle previous work and Methods. Each subsection in Section 2 seems to start with a short introduction and previous work.
- l. 108 ff. The model architecture (especially the processing of the sampling layers) should be described more clearly.
- It is not clear why the authors chose a specific $\beta$ in different places. (For example in Fig. 4; Fig. 8b seems to be cherry-picked for cNVAE, and an especially bad example for VAE)
- All Figures: more descriptive captions would be beneficial.

**Minor**
- The authors could reference to specific sections/figures in the appendix to make navigation easier.
- The layout of the tables does not follow the style guidelines for scientific tables (see NeurIPS guidelines)
- Fig. 4a is not mentioned or described in the text.
- Fig. 4b: mention R^2 score in caption. Write R^2 numbers down also for “bad” models. It is not clear why some bars are “missing”.
- The authors should include a short description of the DCI framework.
- Fig. 6: figure labels are very tiny and hard to read, 6d is so tiny that it is hard to interpret it at all.

**Questions:**

- Which $\beta$ is used for Fig. 3?
- Can the authors reiterate how cNVAE is used for data generation?
- How does cNVAE compare to the non-compressed version (with matched latent dimensions)?
- How do the different metrics perform on the raw data? This would add a simple baseline which could help to interpret the results.
- Fig 4 a: from which layer do the neurons come? Maybe even highlight them in Fig. 3.



**Limitations:**

Yes, very well done.

---

> ### Author Rebuttal · Authors · 2023-08-10
>
> > Evaluation...
>
> We like these suggestions and find some particularly exciting. Although, we consider several of them interesting future directions, given the manuscript already covers a lot of ground (see general response for a summary). In short, the present work is meant as an empirical report of all the cool things that happen to the representations when the latent space is designed in a hierarchical way. As a next step, we were planning to quantify the geometry of the latent spaces using tools from neural population geometry approaches that have recently gained traction.
>
> > the learned representation (manifold)
>
> Motivated by your comment, we performed some first-level analyses to investigate the geometry of representations using the simple method of “effective dimensionality”, which is computed based on eigenvalues of the covariance matrix and provides an estimate of manifold dimensionality. We found that across a broad range of $\beta$ values, the dimensionality of cNVAE representations was substantially larger than that of VAE, suggesting that their representational geometries are ultimately different in a quantifiable way. This result be taken as a starting point for a thorough analysis of the representational geometries. Please see our Figure R1 and global rebuttal for more details.
>
> > the necessary dimension of the latent space
>
> It is interesting to quantify the dependence of our results on the number and organization of latents (i.e., how many latent groups, how many latent variables per group, etc). We did not perform these analyses for the rebuttal due to lack of time, but if the reviewer feels like this would enhance the quality of our paper we will do so and include the results in the final version. Please let us know!
>
> > counterexamples where the model fails and why it fails; further investigation of the receptive fields for different neurons
>
> We will also explore more neurons and find counterexamples to report in the supplementary. We will dig a bit deeper to understand why some neurons are more aligned, while others are not.
>
> > The paper would benefit from some restructuring
>
> Thank you for the suggestion, which we agree with. We will add a “related work” section and clarify some of the missing key background info, which is added to our global rebuttal.
>
> > l. 108 ff. The model architecture (especially the processing of the sampling layers) should be described more clearly.
>
> We apologize for the lack of clarity and will include more details in a substantially revised description of the model architecture and our specific contributions.
>
> > It is not clear why the authors chose a specific beta in different places. (For example in Fig. 4; Fig. 8b seems to be cherry-picked for cNVAE, and an especially bad example for VAE)
>
> Overall, we (and others) find that different choices of the $\beta$ will result in different model properties. As a result, while we scanned across all betas, we displayed results for certain betas to best demonstrate our points. For instance, in Figures 3 and 4 we chose beta values that maximized the overall informativeness score for each architecture in order to make a fair comparison ($\beta = 0.15$ for the cNVAE, $\beta = 1.5$ for the VAE -- please ignore the typo in Figure 4 caption, it should say $\beta=1.5$, we will fix this). You can see that this is the case from Figure 5 where we show DCI scores for all betas. In Figure 8b we deliberately chose a larger beta for VAE because previous work ([Higgins et al. 2021](https://www.nature.com/articles/s41467-021-26751-5)) suggested that increasing $\beta$ values increases alignment, which we did not observe here. In contrast, even a very small $\beta = 0.01$ for cNVAE results in a large alignment. This result (paired with other observations) suggests that alignment, as we measured it here, emerges due to the architecture rather than from large $\beta$ values alone---although we do observe some dependence on $\beta$ values, so ultimately a combination of both architecture and loss is important (but mostly architecture).
>
> > All Figures: more descriptive captions would be beneficial.
>
> This will be fixed in our revised manuscript, including the other minor suggestions (which we thank the reviewer for).
>
> > Which beta is used for Fig. 3?
>
> $\beta = 0.15$ for cNVAE, and $\beta = 1.5$ for VAE.
>
> > Can the authors reiterate how cNVAE is used for data generation?
>
> cNVAE was not used for data generation.
>
> > How does cNVAE compare to the non-compressed version (with matched latent dimensions)?
>
> With a matched number of latents, it is expected that NVAE will severely underperform cNVAE. This is partly because trying to match their latent dimensionality necessitates reducing the number of hierarchical latent groups in the NVAE. From previous work ([Child 2021](https://openreview.net/forum?id=RLRXCV6DbEJ)), we know that the “stochastic depth” of hierarchical VAEs is the key reason behind their effectiveness; therefore, we expected that reduced depth was going to hurt an NVAE with matched # of latents.
>
> To test this, we trained a non-compressed NVAE with roughly the same # of latents (440 vs. 420 for cNVAE), the same number of parameters and conv layers, but necessarily with a reduced number of latent groups (11 vs. 21 for cNVAE). We tested its *untangling* performance (similar to Figure 4b) and found that it dropped significantly compared to cNVAE in predicting every ground truth variable, but it was still higher than VAE. The average untangling scores are as follows:
>
> - cNVAE: **0.898**
> - NVAE: 0.639
> - VAE: 0.406
>
> We thank the reviewer for this question because it prompted us to demonstrate clearly that the latent space of NVAE is unnecessarily and redundantly large. An observation made by others as well ([Hazami et al., 2022](https://arxiv.org/abs/2203.13751)). We are planning to include this new result in Figure 4b, along with performance on raw data (as was also suggested) to further enhance the comparisons.

---

> > ### Comment · Reviewer_Y46x · 2023-08-14
> > **Response to rebuttal**
> >
> > I thank the authors for their thorough responses and I appreciate the additional experiments.
> >
> > Especially the results of the ablation experiments are very interesting, and a good starting point for an in-depth investigation and discussion of different aspects of the model.
> >
> > For example, I still think that investigating different dimensionalities of the latent space could add interesting results to the paper. As for a perfectly disentangling model, the stimulus dimension should already be sufficient.
> >
> > When reading the reviewers’ comments and all responses, I see a lot of promised clarifications and additional experiments. While I know, that it is not possible in the short rebuttal period to run a lot of different experiments, nor is it possible to upload a revised version of the manuscript, this shows that the current manuscript is, if at all, in a borderline state for publication. Therefore, I stand by my initial assessment.

---

> > > ### Author Response · Authors · 2023-08-17
> > >
> > > Thank you for your comments.
> > >
> > > > Especially the results of the ablation experiments are very interesting
> > >
> > > We share the reviewer’s enthusiasm for these experiments. The ablation experiments are one of many ways to follow up on our initial results and are consistent with our goal of demonstrating the utility of our modeling framework—a goal that will enable us to explore all the exciting future research.
> > >
> > > > I still think that investigating different dimensionalities of the latent space could add interesting results to the paper. As for a perfectly disentangling model, the stimulus dimension should already be sufficient.
> > >
> > > We agree and also find this result quite intriguing. While the ground truth is 11-dimensional, we observe dimensionalities of ~50 for the cNVAE latents. What is encoded in those extra dimensions? We speculate that this relates to the nature of non-linear embeddings, where a lower-dimensional but curved manifold can be approximately captured in a higher-dimensional (but still dimension-limited) subspace.
> > >
> > > More broadly, our simulation/modeling framework presented here provides a straightforward instance to understand how hierarchically structured systems can approximate natural laws, which are generally non-linear. Such laws that govern the structure of stimuli in the world typically connect a limited number of ground truth variables to complex, high-dimensional realizations. In our case, the rules of projective geometry determine how the flow frames (raw data), spanning hundreds of dimensions, are derived from the 11 underlying variables.
> > >
> > > In this point of view, the extra latent dimensions can be regarded as effective or *emergent* degrees of freedom that are nontrivial (and likely nonlinear) combinations of the true generative factors. Are those extra dimensions physiologically relevant? One example could be the angle between the *heading direction* and the *gaze direction*. Moving forward, we are excited about investigating the learned latent codes more thoroughly in an attempt to find answers to such questions.
> > >
> > > Nevertheless, a deep dive into exploring the relationship between an 11-dimensional ground truth and an effectively larger dimensional latent code calls for a dedicated, subsequent paper. In our estimation, such an in-depth treatment would be more valuable than hastily incorporating a brief and surface-level analysis into our current paper.
> > >
> > > The present work is meant to describe the framework's main results and is focused on establishing foundational points that must precede the direction that the reviewer suggests. Please also see our latest follow-up comment under the general rebuttal. Thus, while we are in complete agreement that these are interesting directions, we assert that our current figures are necessary to establish this framework, and are important in their own right.
> > >
> > > > When reading the reviewers’ comments and all responses, I see a lot of promised clarifications and additional experiments. While I know, that it is not possible in the short rebuttal period to run a lot of different experiments, nor is it possible to upload a revised version of the manuscript, this shows that the current manuscript is, if at all, in a borderline state for publication. Therefore, I stand by my initial assessment.
> > >
> > > More broadly, the fact that additional analyses would also be interesting should not necessarily detract from the current content of the paper, which is also novel and establishes the foundation for the future directions suggested by the reviewer. We believe that it is an unfair metric to judge a paper by what could be added rather than its current content. Given that we clarify our current work (as the reviewers have given us great direction on), we hope the reviewer would see this work as an advance, with the potential of exciting follow-up work as a positive rather than negative factor in their judgment.

---

### Official Review · Reviewer_Hbpj · 2023-07-06

**Soundness:** 4 excellent
**Presentation:** 4 excellent
**Contribution:** 4 excellent
**Rating:** 7
**Confidence:** 4

**Summary:**

The paper investigates the alignment of representations in deep generative models with activity in mammalian nervous systems. They provide a novel dataset on motion perception (Retinal Optic Flow Learning or "ROFL") against which to test computational models of Helmholtzian analysis-by-synthesis. The dataset generates retinal flows based on disentangled latent factors determining the motion and appearance of objects.  The paper then tests a novel extension of the deeply hierarchical Nouveau VAE which imposes a "pyramidal" scaling of latent spaces through the hierarchy for its ability to capture the retinal-flow dataset in a "Helmholtzian" way, as well as its alignment with a pre-published macaque electrophysiology dataset on the middle-temporal visual area (MT).

**Strengths:**

The paper presents an original deep generative model architecture, trainable with the typical ELBO or $\beta$-VAE objective by black-box variational inference, which captures a desirable and sought-after feature of generative modeling in the brain.  The experimental evaluation compares against maximum-likelihood/reconstruction-loss estimation and a deterministic autoencoder, a nice comparison for making sure that the neural data aligns best with a deep probabilistic generative model as opposed to a non-probabilistic deep neural network.

**Weaknesses:**

As the authors admit in the Discussion section, they trained their compressed Nouveau VAE (cNVAE) on optical flow data rather than on video/pixel data.  While this does map better to the MT area than pixels do, a very deep model ought to be able to learn to represent optic flow in the course of predicting pixels.

While this supervision with feature-engineered optical flow data in place of learning from time-series does remain a weakness, the authors have presented a sufficiently interesting additional contribution in addressing my question below that I will be raising my score.

**Questions:**

Do the authors have a plan to apply their cNVAE in a "clockwork VAE" setting to predict videos?  Can we see if optic flow emerges from training a deep generative model?  Moreover, is there an ablation study able to show how the recognition model $q$ versus the generative model $p$ in the cNVAE contribute to its alignment with neural representations?

The authors have presented interesting ablation studies in their rebuttal, whose results make perfect sense in retrospect and yet which I could not have predicted a priori.  I am thus convinced I need to raise my score.

**Limitations:**

The authors have addressed potential limitations and impacts.

---

> ### Author Rebuttal · Authors · 2023-08-10
>
> > As the authors admit in the Discussion section, they trained their compressed Nouveau VAE (cNVAE) on optical flow data rather than on video/pixel data.
>
> We appreciate this point, and it relates fundamentally to key choices in our approach. The raw visual image (photons on the retina) is processed by many layers of the visual pathway through many specialized computations throughout the retina and cortex, and these transformations are the subject of much computational work (e.g., [Nishimoto & Gallant](https://www.jneurosci.org/content/31/41/14551)). Here, we wanted to simplify the stimuli to focus on a clearly defined and focused question of optic flow: starting with the optic flow (velocity field resulting from movement in the scene and by the observer/eyes) to test if our artificial model can extract the generative “ground truth” variables from the optic flow stimulus. This stimulus also purposefully matches a majority of experiments in area MT that use moving dots, including the dataset in our study.
>
> The use of optic flow instead of raw-pixel stimuli of course allows the cNVAE to focus on the problem of inferring a representation of the generated motion fields: and, in this sense, not need to simultaneously solve the difficult problem associated with extracting motion fields from pixel data (and/or the conjoint problem). In short, our manuscript focuses on this problem – and please see our summary of the general goals of this manuscript – and, in achieving high performance on explaining MT data, implies that MT may be doing something similar (when given equivalent dot-motion stimuli).
>
> In terms of solving the harder problem of processing raw spatiotemporal movies, we propose our approach provides a new possible path for this and plan to extend our cNVAE to operate on raw images in the future.
>
> > While this does map better to the MT area than pixels do, a very deep model ought to be able to learn to represent optic flow in the course of predicting pixels.
>
> Yes, in the future we will test this. We expect such solutions based on raw pixel data will be achievable with the same framework and more layers of latent variables, although it is possible that these problems are linked.
>
> > Do the authors have a plan to apply their cNVAE in a "clockwork VAE" setting to predict videos? Can we see if optic flow emerges from training a deep generative model?
>
> Thank you for this question. In the Clockwork VAE paper, they use hierarchical latents but in the temporal domain and report the benefits of this inductive bias. In our study, we decoupled spatial and temporal aspects of motion processing and focused solely on the spatial hierarchy. This simplification allowed us to demonstrate that hierarchical models were able to understand multiscale data (like the real world) where a non-hierarchical VAE struggled.
>
> It would be interesting to add temporal hierarchical structure to ROFL in a spatiotemporal video setting and use it to investigate how latents would capture coexisting fast and slow dynamics. For instance, one can include objects that move much faster or slower than self-motion. In the clockwork VAE paper, they found that higher latents captured objects with slower time scales. Here, we found that higher latents specialize in encoding object-related features. It would be interesting to combine their setting with our architecture to find out what happens in the spatiotemporal domain and whether the model learns brain-like representations. This is a scientifically relevant and important question because there is also a hierarchy of intrinsic time scales in cortical dynamics ([Murray et al., 2014](https://www.nature.com/articles/nn.3862)). However, we believe such an important problem deserves its own full treatment in a separate, follow-up paper.
>
> > Moreover, is there an ablation study able to show how the recognition model q versus the generative model p in the cNVAE contribute to its alignment with neural representations?
>
> Thank you for this excellent suggestion. We performed ablation experiments and found complementary insights into why cNVAE outperforms alternative models. We report these results in Figure R1, along with a brief discussion and interpretations in our global rebuttal. Overall, we found your suggestion to be very interesting because understanding the contributions from bottom-up and top-down connections in the cortex is a central problem in neuroscience. We are planning to tackle this question more systematically in the future.
>
> > The authors do not appear to include a Broader Impacts section to review.
>
> We had to fill it out here on OpenReview rather than having a designated section in the paper. Our response is copied below for the reviewer’s consideration:
>
> "**Broader Impacts**: We introduce a new simulation framework that facilitates hypothesis generation and testing in science, but it is not sophisticated enough for potentially harmful applications. Thus we do not anticipate negative social impacts from this work."

---

### Author Rebuttal · Authors · 2023-08-10

Here we address the most common concerns and highlight additional analyses inspired by them. We realize our work did not come across clearly to all reviewers and we offer a brief summary of our main points first:

We were motivated by the idea that representations of the natural sensory world involve the inference of the underlying causes of the senses. This is foundational in theories of perception, that posit that our brains learn hierarchical generative models of the world. We focused on the role of hierarchical inference in the learned representations of natural motion and its causes (moving objects and self-motion). Our paper’s main points are:

(i) We created a simulation framework (ROFL) for synthesizing motion stimuli. ROFL enables control over ecologically relevant factors (self-motion and objects) while avoiding confounds due to texture. Importantly, ROFL has a hierarchy of spatial scales — just like the real world — which interact in nontrivial ways (see Figs. 1c-d).

(ii) We introduced compressed NVAE (cNVAE) that greatly reduced the number of latents.

(iii) We evaluated the representations of hierarchical and non-hierarchical models using a multitude of metrics and found the cNVAE performed favorably.

(iv) We measured neural prediction and alignment of the cNVAE and comparison models using recordings from area MT, where we outperform the SOTA by a factor of 2.

In sum, we focused on understanding the relationship between the learned representations (VAE latents) and “ground truth” causes of sensor data. We showed how representations depend on architecture (hierarchical vs. non-hierarchical), loss (variational vs. maximum likelihood), and the training set.

We will add a Related Work section to the manuscript in the appendix, which should further clarify the contributions of the manuscript. As we are limited to 6K characters, we have included an abridged draft below.

# Related work (draft)
## Neuroscience and VAEs
The connection between VAEs and neuroscience is reviewed by (Marino 2022), but direct comparisons have been limited to these recent papers:

Higgins 2021: Trained beta-VAE models on face images and found that beta-VAE discovers individual latents that are aligned with IT neurons.

Storrs 2021: Trained PixelVAE on synthesized images of glossy surfaces and found that the model spontaneously learned to cluster images according to underlying physical factors and mimicked human perceptual judgments.

Csikor 2022: Investigates the properties of representations and inference in a biologically inspired hierarchical VAE called Topdown VAE that captures key properties of representations in V1 and V2 of the visual cortex.

Miliotou 2023: learned mappings from the latent space of a hierarchical VAE to fMRI voxel space supported improved reconstruction of images from brain data. Ablation experiments find hierarchy is an essential component.

## Hierarchical VAEs
Ladder VAE (LVAE) was the first to introduce hierarchy to VAEs, which improved upon standard VAEs by sharing information between the inference and generative networks, allowing LVAEs to learn deeper representations. Building on LVAE, Maaløe et al. 2019 introduced Bidirectional-Inference Variational Autoencoder (BIVA), using skip-connections to further enhance the flow of information among latent variables. Both the LVAE and BIVA enabled VAEs to effectively leverage deep stochastic hierarchies.

Recently, the hierarchical Nouveau VAE (NVAE) (Vahdat & Katz, 2020) achieved SOTA in several benchmarks and generated high-quality faces. Very Deep VAE (vdvae; Child 2021) achieved impressive performance on complex image benchmarks. Neither work evaluated how the hierarchical latent structure changed the quality of learned representations. As far as we know, ours is the first study focused on the evaluation of representations in hierarchical VAEs with applications to neuroscience data.

Additionally, NVAE and vdvae have an undesirable property: their convolutional latents result in a latent space that is several orders of magnitude larger than the input space defeating the main purpose of autoencoders. For vdvae, a tiny subset (only 3%) of its latent space is necessary [Hazami 2022](https://arxiv.org/abs/2203.13751). We demonstrate that it is possible to compress hierarchical VAEs and focus on investigating the latents.

## Evaluating DNNs on predicting biological neurons.
Many studies have evaluated DNNs for predicting brain responses, but most are on static image processing (“ventral stream”), most notably. In contrast, motion processing in the dorsal stream has only been considered thus far in Mineault (2021), who used a deep neural network to extract ground truth variables from simulated drone flight ("AirSim"). They evaluate neural prediction across many areas and achieved SOTA in prediction for the dataset we consider here, although we greatly outperform them.


# New rebuttal results, future work
## Ablation experiments
We performed ablation experiments and found that this offers insight into what is represented by the latents in cNVAE. Lesioning top latents in the encoder pathway removes the object from the reconstructed output, leaving an unperturbed background (see our rebuttal Figure R1-a). In contrast, bottom latent lesions disrupt self-motion and leave object position and velocity mostly undisturbed (Fig R1-d). We extended these ablations to our MT neuron predictions. The performance drops most dramatically when the bottom latents are disrupted. Lesioning the decoder did not lead to significant performance drops (Figure R1-e; top).

## Manifold analysis
We performed a manifold analysis using effective dimensionality (ED; see Figure R1 for a definition) and found ED is larger for cNVAE compared to VAE even though both models have the same number of latents and achieve almost identical loss (Fig. S5). This suggests a future direction to understand how cNVAE accomplishes these properties.

---

> ### Author Response · Authors · 2023-08-17
>
> We are grateful for the engagement of the reviewers and their insightful critiques and comments. It goes without saying, the degree that they have expressed their reservations about our paper allows us to potentially address/explain (in our rebuttals), and these reviewer comments have contributed to a much improved revised manuscript.
>
> We would like to restate that this manuscript presents a novel simulation framework for unsupervised training and a novel compressed hierarchical VAE that is able to infer the structure of the world from unsupervised learning. We furthermore demonstrate through a wide range of metrics currently in the field that it outperforms SOTA across many of them. As our individual reviewer rebuttals discuss, the simplicity and control allowed in the ROFL framework will facilitate many future directions, including the nature of nonlinear embeddings, the role of dimensionality, and how hierarchical models can capture simple laws (like projective geometry of ROFL) that they are not explicitly made for.
>
> While we clearly share interests with the reviewers – which is one reason we are excited about this foundational work – the present manuscript can only include so many topics. We consider this work the first in what we plan to be a series of papers building upon the established framework, each going into details of the promising additional results revealed by our rebuttal figure. [Based on some reviewer comments, we might have even decreased the span of our results further to enable more detail in the results that we do present, but feel (with the revisions) we now have a good balance.]
>
> In considering whether our current scope is broad enough, we would like to point out a recent review paper ([Doerig et al., 2023](https://www.nature.com/articles/s41583-023-00705-w)), which describes the necessary ingredients for using artificial neural networks (ANN) as a computational language for expressing falsifiable theories about brain computation.
>
> Based on the criteria of Doerig et al., our work contributes to 3 out of 4 core ingredients of the *neuroconnectionist* research programme. Namely, ***objective functions*** (VAE vs. AE loss; see Table 2 in the main paper), ***training data sets*** (ROFL categories with different components/structure; Tables 1), and ***architectures*** (hierarchical vs. non-hierarchical; Table 2). The only core ingredient left out of our study is the ***learning rule***, for which we used backpropagation with stochastic gradient descent.
>
> In this sense, we respectfully maintain that the results of our paper stand on their own, without the need to add additional work, as interesting as it might be.

---

### Decision · Program_Chairs · 2023-09-21

**Decision:**

Accept (poster)

**Comment:**

This paper examines whether hierarchical Bayesian inference could provide an account of motion processing in the primate brain. The authors develop a synthetic dataset for training networks on optical flow data, and examine the representations formed in various types of variational autoencoders. The authors show that hierarchical VAEs trained on this dataset develop representations that untangle factors of variation, are disentangled, and align well to real neuron responses in monkey MT in response to moving dot stimuli.

This was a challenging decision, as the scores were borderline. The critical reviewers were concerned that the paper failed to really provide insight into what the key factors were for model alignment, that the hierarchy didn't actually seem to be all that important for alignment to the neural data, and that the methods for assessing the model were circular. The authors provided extensive rebuttal with new data and many changes proposed. In discussion, the reviewers were then also concerned that there were too many changes promised, and the paper might change too much in the interim.

As an AC, after reading through the paper, further considering the reviews, and looking at the authors' rebuttals, I ended up coming down in favour of acceptance. The central reason I ended up deciding on accept was that the paper does present an interesting new synthetic dataset, and the fact that a hierarchical VAE trained on this dataset matches responses in primate MT to moving dot stimuli is interesting and would seem to support a theory of Bayesian inference accounting for the responses in the dorsal stream. As well, the authors did work hard to address the reviewer concerns.

However, I also want to caution the authors on a couple of points and encourage them to highlight these limitations (as well as making all the changes they promised):

1. The reviewer's concern that the non-hierarchical model also accounts for MT responses well is quite valid. That would seem to suggest that *hierarchical* inference is not actually all that important for explaining the responses, just inference. In rebuttal, the authors suggest that predictive power alone shouldn't be weighed too heavily. Which may be true, but the fact remains that predictive power isn't *meaningless*, so the authors should recognize this explicitly as a complication for their conclusions.

2. When comparing to the Mineault et al. (2021) paper, the authors recognize that one big limitation of their study relative to this previous one is the fact that they used a model that can only receive moving dot stimuli. Indeed, the R numbers they report for MT for the model in Mineault et al. (2021) are therefore only the numbers for one dataset explored in that work, but their model actually achieved a much higher R value (.381) for a video dataset. This should be explicitly mentioned in the discussion section where these issues are touched on, i.e., it's only a 2x improvement for the dots stimuli, and we can't assess the performance on the video stimuli due to the modelling decisions here. It is possible that a hierarchical VAE trained and tested on video stimuli would align more poorly than the model from that previous work.